# Quantification of purified endogenous miRNAs with high sensitivity and specificity

Soochul Shin[1,7], Yoonseok Jung[2,7], Heesoo Uhm [1], Minseok Song[1], Soomin Son[2,3], Jiyoung Goo[4,5], Cherlhyun Jeong [4,5], Ji-Joon Song [6], V. Narry Kim [2,3] & Sungchul Hohng [1✉]

MicroRNAs (miRNAs) are short (19–24 nt) non-coding RNAs that suppress the expression of protein coding genes at the post-transcriptional level. Differential expression profiles of miRNAs across a range of diseases have emerged as powerful biomarkers, making a reliable yet rapid profiling technique for miRNAs potentially essential in clinics. Here, we report an amplification-free multi-color single-molecule imaging technique that can profile purified endogenous miRNAs with high sensitivity, specificity, and reliability. Compared to previously reported techniques, our technique can discriminate single base mismatches and single-nucleotide 3'-tailing with low false positive rates regardless of their positions on miRNA. By preloading probes in *Thermus thermophilus* Argonaute (*Tt*Ago), miRNAs detection speed is accelerated by more than 20 times. Finally, by utilizing the well-conserved linearity between single-molecule spot numbers and the target miRNA concentrations, the absolute average copy numbers of endogenous miRNA species in a single cell can be estimated. Thus our technique, Ago-FISH (Argonaute-based Fluorescence In Situ Hybridization), provides a reliable way to accurately profile various endogenous miRNAs on a single miRNA sensing chip.

[1] Department of Physics and Astronomy, and Institute of Applied Physics, Seoul National University, Seoul, Republic of Korea. [2] Center for RNA Research, Institute for Basic Science (IBS), Seoul, Republic of Korea. [3] School of Biological Sciences, Seoul National University, Seoul, Republic of Korea. [4] Center for Theragnosis, Korea Institute of Science and Technology, Seoul, Republic of Korea. [5] KHU-KIST Department of Converging Science and Technology, Kyunghee University, Seoul, Republic of Korea. [6] Department of Biological Sciences, Korea Advanced Institute of Science and Technology (KAIST), Daejeon, Republic of Korea. [7] These authors contributed equally: Soochul Shin, Yoonseok Jung. ✉email: shohng@snu.ac.kr

MicroRNAs (miRNAs) are short regulatory RNAs that control the expression of protein-coding genes at the posttranscriptional level[1,2]. The dysregulations of miRNAs are related to cellular malfunctions and numerous diseases[3]. For instance, a large number of miRNAs are found to be dysregulated in a broad spectrum of cancers in a disease-specific fashion[4], and specific miRNAs circulating in the blood of numerous cancer patients are found to be highly abundant, and remarkably stable compared to those in the blood of healthy individuals[5], making miRNAs ideal tumor markers for early detection. Recently, it has been reported that certain viruses, such as SARS-CoV (severe acute respiratory syndrome coronavirus) express their own miRNAs, or influence the expression of cellular miRNAs[6–8], suggesting a potential use of miRNA detection as a biomarker for viral infection.

The small size and low abundance of miRNAs present special challenges that make conventional techniques to detect larger RNAs inadequate for miRNA detection. In addition, many miRNA families are highly homologous and differ by only a single base[9], highlighting the need of miRNA detection with high specificity. A number of miRNA detection techniques are currently commercialized, but they have their own limitations[10–14]. In a conventional microarray analysis, extracted miRNAs are poly (A)-tailed, ligated with a biotinylated DNA strand, and captured on an array of spots of a microarray plate, each of which are coated with specific DNA strands complementary to the target miRNA. Binding of miRNAs on the spot is detected by adding fluorescently labeled streptavidin. Using this technique, we can monitor a large number of target miRNAs in low cost, but the specificity and dynamic range of microarray analysis are limited because a RNA–DNA duplex of 20 base pairs is quite stable, and as a result, single base pair difference cannot be detected. In a conventional miRNA sequencing, extracted miRNAs are ligated with RNA adapters on both the 3′- and 5′-sides, converted into DNA strands by reverse transcriptase, and amplified using polymerase chain reaction (PCR). After each enzymatic step, the reaction products should be purified using PAGE (polyacrylamide gel electrophoresis), and the final products are sequenced using a sequencing machine. The RNA-sequencing technique is suitable for high-throughput miRNA profiling, and can discover uncharacterized miRNAs, but rather expensive and its precision of quantitation is poor. Furthermore, uneven enzymatic reactions cause bias in miRNA detection. Recently, the bias problem of RNA sequencing was improved by adopting random adapters and crowding reaction conditions, but not completely solved[15]. In a conventional quantitative real-time PCR (qRT-PCR), extracted miRNAs are converted into DNA using reverse transcription, and progress of quantitative PCR is monitored using a special fluorescent probe. qRT-PCR is known to be relatively precise, but prone to bias and can be easily contaminated by pri- and pre-miRNAs[13,16].

Here, we report a multicolor single-molecule imaging technique, Argonaute-based fluorescence in situ hybridization (Ago-FISH), that can sequentially profile multiple purified endogenous miRNA with high sensitivity, specificity, and reliability.

## Results

### Acceleration of target binding/dissociation.
The key ideas of Ago-FISH are explained in Fig. 1. We first prepared a target synthetic RNA that contained let-7a target sequence on the 5′-side and U15 linker on the 3′-side (Fig. 1a). The RNA was biotinylated on the 3′-end, and thus could be immobilized on a quartz surface by using streptavidin–biotin interaction. For miRNA detection, we prepared three DNA probes (let-7a_PS, let-7a_PM, and let-7a_PT in Fig. 1b) to detect the seed, mid, and tail

regions of let-7a miRNA, respectively; the sequences colored in black, green, and red are complementary to the regions of let-7a miRNA indicated by the bars of the same color in Fig. 1a. Since the DNA probes were distinctly labeled with Cy7, Cy3, and Cy5, their binding to miRNAs could be distinguished using single-molecule multicolor fluorescence imaging. When we designed DNA probes, the lengths of the complementary part of the DNA probes were optimized so that they had comparable binding/dissociation kinetics on target miRNAs by compensating the varying GC contents of the seed, mid, and tail regions. The sequence of noncomplementary part of DNA probes was not critical, but selected for DNA probes not to form secondary structures. The overall length of DNA probes was maintained at 21 nt.

In miRNA detection using bare DNA probes[17], the target binding of DNA probes is an slow process with a binding rate ~1000 s/nM (refs. [18,19]). To solve this problem, and accelerate the target binding of DNA probes, we preloaded the DNA probes in *Thermus thermophilus* Argonaute (*Tt*Ago). *Tt*Ago was selected among various Argonaute proteins from different species, because it was well characterized in the laboratory, stable, and did not have a cleavage activity at the temperature used for the experiments (30 °C)[20,21]. It is possible that a better choice of Argonaute proteins exists for Ago-FISH. After assembling *Tt*Ago and DNA probes, we simultaneously injected the three kinds of *Tt*Ago-loaded DNA probes into a detection chamber where target RNAs were immobilized on a polymer-coated quartz surface using streptavidin–biotin interaction (Fig. 1c), and binding of the three DNA probes to target miRNAs were monitored using a multicolor single-molecule fluorescence microscope[22–24]. Following the injection of DNA probes, a large number of single-molecule spots appeared in all the imaging channels of Cy3, Cy5, and Cy7 (Fig. 1d). On the other hand, a negligible number of single-molecule spots were observed without target miRNA immobilization (Fig. 1e), and the number of single-molecule spots were significantly reduced in the presence of ten times excessive unlabeled competitor DNA probes (Fig. 1f). As shown in representative fluorescence time traces (Fig. 1g), *Tt*Ago-loaded DNA probes exhibited significantly increased binding rates of DNA probes compared to bare DNA probes; quantitative measurements showed more than an order of magnitude increase (19.7, 20.1, and 24.1 times for let-7a_PS, let-7a_PM, and let-7a_PT, respectively) of the binding rate constants of the DNA probes (Fig. 1h). Interestingly and favorably for miRNA detection, the dissociation rates were also increased (3.2, 1.8, and 2.4 times for let-7a_PS, let-7a_PM, and let-7a_PT, respectively) when the DNA probes were loaded in *Tt*Ago (Fig. 1i).

### High specificity for the whole region of target miRNAs.
For accurate miRNA profiling, miRNA detection scheme needs to have a high specificity for the whole region of miRNA. In the miRNA detection technique where miRNAs were captured using a strong RNA–LNA hybridization of the 3′-half of miRNAs[17], the high specificity of single-nucleotide discrimination could not be demonstrated for the miRNA region used for surface immobilization. To demonstrate the high specificity of Ago-FISH for the whole region of miRNA, we prepared three let-7a mutants—let-7a_m1, let-7a_m2, and let-7a_m3—that had a one-nucleotide mutation in the seed, mid, and tail regions of let-7a, respectively (Fig. 2a). Representative binding traces of the let-7a probes on the let-7a mutants clearly showed that the binding of the DNA probes targeting the mutated region was significantly and selectively decreased (Fig. 2b). For a quantitative analysis, we selected all time traces which showed the binding of any of the three DNA probes during observation, and obtained the average duty cycle

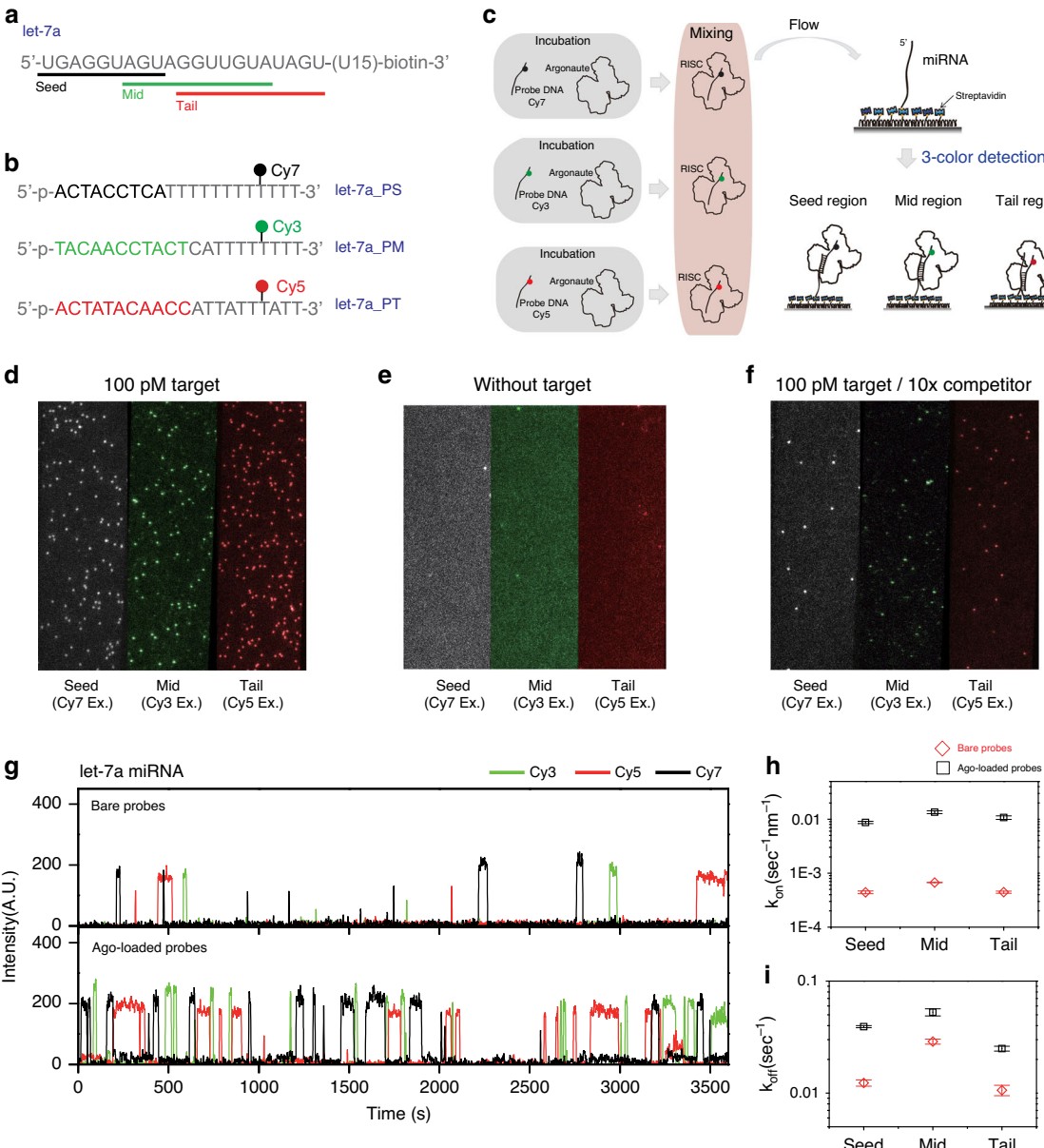

**Fig. 1 miRNA detection scheme. a** Design of the target RNA containing let-7a sequence on the 5′-side, U15 linker on the 3′-side, and biotin at the 3′-end. **b** Design of DNA probes targeting let-7a. let-7a_PS, let-7a_PM, and let-7a_PT targeting the seed, mid, and tail regions of let-7a, respectively, are labeled with different fluorophores as depicted. **c** Scheme to accelerate miRNA detection. DNA probes are loaded in *Tt*Ago, and their biding to surface-immobilized target miRNAs is monitored using multicolor single-molecule fluorescence imaging. **d** Single-molecule spots of 100 pM let-7a. **e** Single-molecule spots without target let-7a. **f** Single-molecule spots of 100 pM let-7a in the presence of ten times excess unlabeled DNA probes. **g** Representative fluorescence intensity time traces that show binding of bare DNA probes (top) and *Tt*Ago-loaded DNA probes (bottom) on surface-immobilized target let-7a miRNAs. **h** Comparison of the binding rate constants ($k_{on}$) of bare DNA probes (red) and *Tt*Ago-loaded DNA probes (black). **i** Comparison of dissociation rate constants ($k_{off}$) of bare DNA probes (red) and *Tt*Ago-loaded DNA probes (black). To obtain the each data point of **h** and **i**, three or more independent experiments were conducted and at least 300 molecules in total were analyzed. Data are mean ± SEM. To obtain $k_{on}$ and $k_{off}$ of Ago-loaded DNA probes, 10 min imaging time was used. To obtain $k_{on}$ and $k_{off}$ of bare DNA probes, 1 h imaging time was used. Source data are available in the Source data file.

(the fraction of observation time in which a probe was binding to the target miRNA) of each probe. As shown in Fig. 2c, the average duty cycle of the mutated region-targeting probes decreased significantly compared to the original values of 19%, 17%, and 23% of let-7a_PS, let-7a_PM, and let-7a_PT, respectively (Fig. 2c). It was noticeable that the duty cycle of let-7a_PT was also reduced a little in the case of let-7a_m2 because the 3′-end of the target recognition sequence of let-7a_PT probe overlapped the mutated base of let-7a_m2. This reduction, however, did not cause any problem but only favorable effect in target discrimination. In the

case of wild-type let-7a, most of traces (98.2%) showed the binding of the three DNA probes except the minor portion of traces, which showed either one-probe binding (0.5 %) or two-probe binding (1.3 %), demonstrating that Ago-FISH could detect target miRNAs with a high fidelity.

To benchmark Ago-FISH against commercialized qRT-PCR techniques, we chose 5% duty cycle of probe binding as a threshold (i.e., we considered a molecule as positive when the duty cycles of all of the three DNA probes were larger than 5%) to identify as a proper target miRNA in Ago-FISH, and quantified

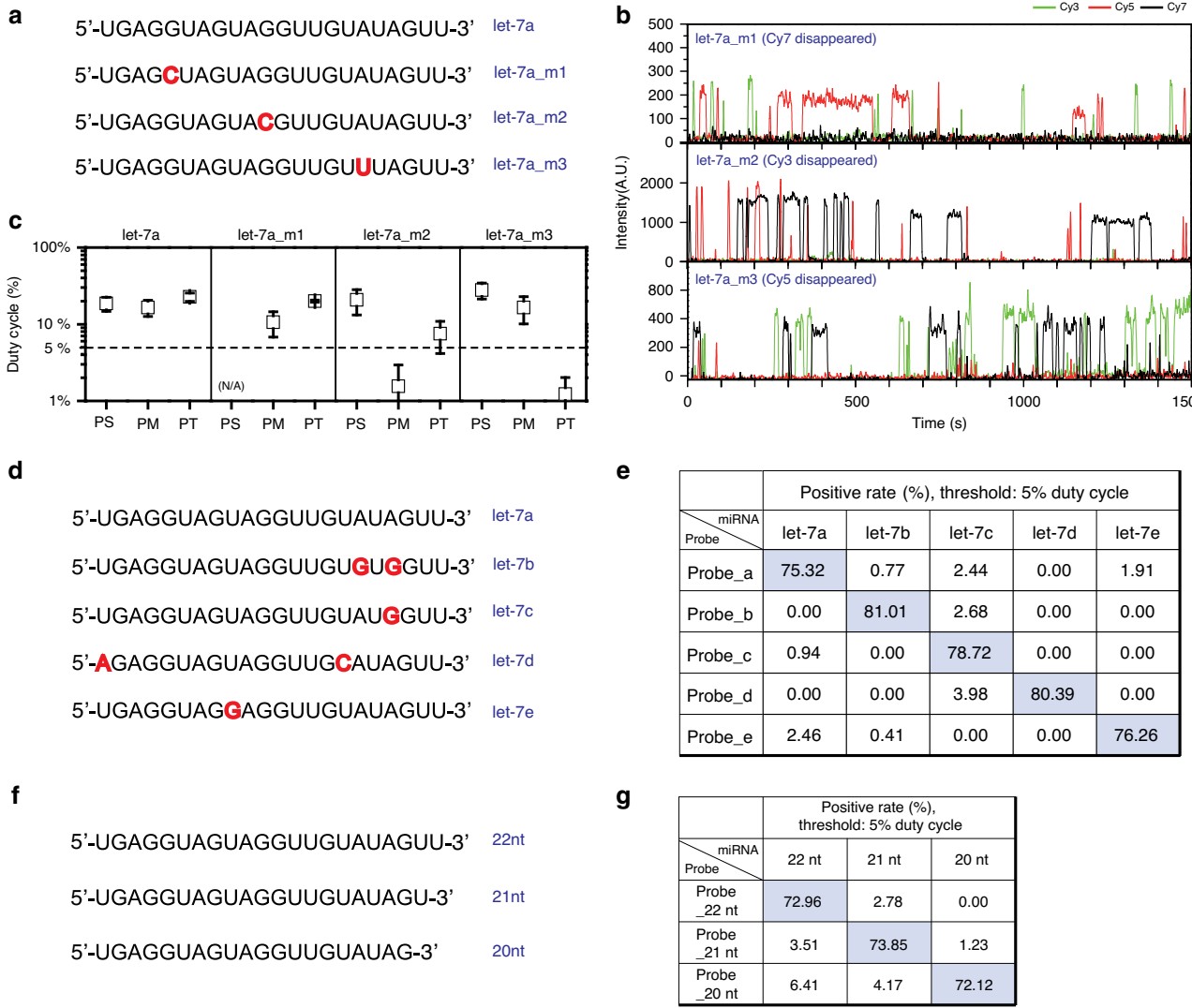

**Fig. 2 miRNA sensing with high specificity for the whole region of miRNA. a** Sequences of let-7a and its single-nucleotide mutations (red) at seed, mid, and tail regions. **b** Representative fluorescence intensity time traces showing binding of let-7a probes to let-7a_m1 (top), let-7a_m2 (middle), and let-7a_m3 (bottom), respectively. **c** Duty cycles of let-7a probes on wild-type let-7a and its mutants. **d** The sequences of let-7 family miRNAs. **e** Positive rates (either true: diagonal, or false: off-diagonal) of let-7 family miRNAs measured using Ago-FISIH. **f** The sequences of let-7 miRNAs with varying U-tail length. **g** Positive rates (either true: diagonal, or false: off-diagonal) of let-7 miRNAs with varying U-tail length measured using Ago-FISH. To obtain each data point of **c**, **e**, and **g**, three or more independent experiments were conducted and at least 300 molecules in total were analyzed. The criteria of 5% duty cycle was used to make the tables. Data are mean ± SEM. For all statistical analyses of miRNA detection in this work, 20 min imaging time was used. Source data are available in the Source data file.

the true and false positive rates of let-7 family miRNAs (Fig. 2d), using DNA probes optimized for their detection (Supplementary Fig. 1). The false positive rates were significantly reduced in Ago-FISH (Fig. 2e) compared to those of commercial qRT-PCR techniques (Supplementary Fig. 2)[13]. A different approach to estimate a false positive rate is to compare the average spot numbers of let-7 family miRNAs (relative detection), in which the true positive rate is automatically assumed as 100% as in qRT-PCR. This approach also revealed improved false positive rates of Ago-FISH compared to those of qRT-PCR (Supplementary Fig. 3).

During maturation, the 3′-end of miRNA can be modified by terminal nucleotidyl transferases, and this RNA tailing affects downstream processing and miRNA stability[25–27]. We tested whether Ago-FISH could reliably discriminate mono- and bi-uridylation of miRNAs, using let-7a miRNAs with different tail regions (Fig. 2f) and DNA probes optimized for their detection

(Supplementary Fig. 1). Figure 2g shows that Ago-FISH can distinguish mono- and bi-uridylated miRNAs with high specificity. In Fig. 2e, g, we arbitrarily chose 5% as the duty cycle threshold. When a better specificity was required, however, the higher threshold value could be used by sacrificing the detection efficiency (defined as the ratio of the true positive to the sum of the true positive and the false negative; Supplementary Figs. 3–5).

**Accurate quantification of purified endogenous miRNAs.** To use Ago-FISH for the quantification of purified endogenous miRNAs, we developed a scheme to immobilize multiple endogenous RNAs on a single-miRNA sensing chip, as depicted in Fig. 3a. We first purified whole endogenous RNAs from cell lysate, and then added adenosine tails to 3′-end of the purified RNAs, using yeast poly(A) polymerase. The tailed RNAs were hybridized with biotinylated poly(T) DNA, captured on a miRNA sensing chip using streptavidin–biotin interaction, and finally

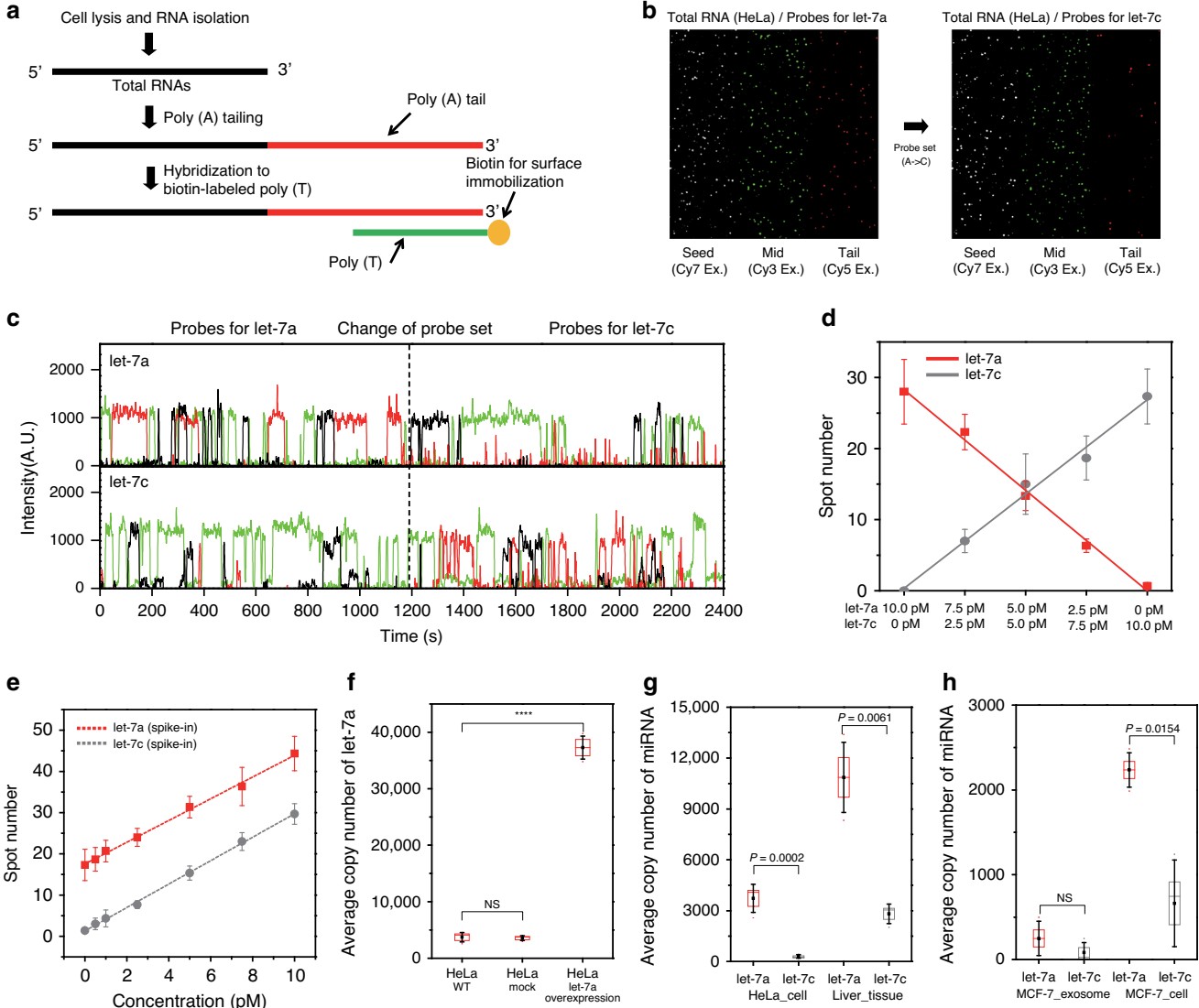

**Fig. 3 Quantification of endogenous miRNAs. a** Scheme to immobilize endogenous miRNAs. Total RNA is purified from cell lysate, tailed with poly(A) at the 3′-end, hybridized with complementary DNA strands, and immobilized on a surface using streptavidin–biotin interaction. **b** Single-molecule images of let-7a and let-7c miRNA from HeLa cell. **c** Sequential detection of endogenous let-7a and let-7c. Representative fluorescence intensity time traces identified as let-7a (top) and let-7c (bottom) are shown. Probes for let-7a detection were exchanged with probes for let-7c detection at 1200 s (dashed line). **d** Calibration for miRNA quantification. The number of spots identified as let-7a miRNA (red) and let-7c (gray) are plotted at varying combination of concentrations of target RNAs. The data are nicely fitted to a linear function (solid lines) with zero $y$-intercept and slopes of 2.8 spot/pM for let-7a and 2.7 spot/pM for let-7c ($R^2 = 0.998$ for let-7a, and 0.996 for let-7c). **e** miRNA spike-in experiments. Varying amount of synthetic let-7a and let-7c miRNAs were spiked in total RNA of HeLa Cell, and their spot numbers were plotted as a function of their concentration. The data are nicely fitted to a linear function (solid lines) with the same slope (2.8 spot/pM), but different $y$-intercepts ($R^2 = 0.996$ for let-7a, and 0.999 for let-7c). **f** let-7a detection in HeLa cell, let-7a overexpression HeLa cell, and mock HeLa cell. **g** Comparison of average copy numbers of let-7a and let-7c per 20 pg total RNA from HeLa cell, and human liver tissue. **h** Comparison of average copy numbers of let-7a and let-7c per 20 pg total RNA from MCF-7 exosome, and MCF-7 cell. To obtain each data point of **d**–**h**, three or more independent experiments were conducted. Data are mean ± SD. $P$ values were determined by two-sided $t$ test (****$P < 0.0001$). Source data are available in the Source data file.

detected using single-molecule multicolor imaging, as described above. In our scheme, all endogenous miRNAs were immobilized on a single-miRNA sensing chip, and therefore different miRNAs could be sequentially detected by simply exchanging DNA probes used for miRNA detection. Figure 3b shows example single-molecule images when let-7a (left panel) and let-7c (right panel) probes were added to the same miRNA sensing chip, where RNAs from HeLa cell were captured. Images of probes targeting the seed (Cy7) and mid (Cy3) regions looked similar between the two panels, whereas the probe images targeting the tail region (Cy5) looked different. Figure 3c shows example fluorescence time

traces that were identified as let-7a (top) and let-7c (bottom), when the initial let-7a probes were exchanged with let-7c probes. When we first prepared endogenous RNAs, we tried to additionally purify short RNAs from total RNAs. However, the addition of the short RNA purification step did not produce any improvement in miRNA detection, but only problems (the purification yield varied from trial to trial, and miRNAs were lost during the additional purification step), and therefore was not used in this work.

To characterize the reliability of miRNAs quantification using Ago-FISH, synthetic let-7a and let-7c miRNAs were tailed with

poly(A) as described above, and their mixture with varying combinations of concentrations was immobilized on a surface. The average number of spots per imaging area (20 μm × 60 μm) identified as let-7a and let-7c, using 5% duty cycle threshold increased linearly with their concentrations during the immobilization step, giving the similar slopes of 2.8 spot/pM for let-7a and 2.7 spot/pM for let-7c (Fig. 3d). On the other hand, the fitting curves for let-7a and let-7c quantification using qRT-PCR were quite different (Supplementary Fig. 6a), indicating that different quantification parameters should be used for the quantification of different miRNAs in the qRT-PCR method. In our scheme of miRNA detection, all different miRNAs are captured using the same poly(A) tail, and therefore similar capturing efficiencies of different miRNAs are naturally expected. However, the miRNA detection efficiencies of DNA probes depend on the design of DNA probes. It remains to be tested whether the similar calibration parameters can be obtained for other miRNA targets by optimizing DNA probes.

To test the same calibration parameters can be used for endogenous miRNAs and synthetic miRNAs, we performed spike-in experiments. Even when the synthetic let-7a and let-7c miRNAs were spiked in total RNA extract of HeLa cell, where various endogenous miRNAs coexisted, no apparent change in the slopes of the fitting curves was observed (2.8 spot/pM for the both miRNAs; Fig. 3e), indicating that the same calibration parameter could be used for the quantification of both synthetic, and endogenous miRNAs in Ago-FISH. The $y$-intercept of the let-7a data, however, was 13 times larger than that of let-7c data (17.5 vs. 1.3, Fig. 3e), indicating let-7a was more abundant than let-7c in HeLa cell, which was consistent with the RNA-sequencing results (data not shown). On the contrary, the qRT-PCR results of the same spike-in experiments showed that fitting parameters for miRNA quantification changed a lot in the presence of endogenous RNAs (Supplementary Fig. 6b), and the Ct (cycle threshold) value of let-7a was measured larger than that of let-7c (Supplementary Fig. 6b), although the absolute quantity of let-7a was larger than that of let-7c due to the endogenous miRNA contribution. Furthermore, the Ct values of let-7a and let-7c of the spike-in experiments were measured larger than those of pure synthetic RNAs (Supplementary Fig. 6) despite of the existence of endogenous let-7a and let-7c miRNAs. We suspected that various endogenous RNAs existing in cell lysate significantly interfered the qRT-PCR quantification of miRNAs.

By using the well-conserved linearity between the target miRNA concentration and the number of spots detected on miRNA sensing chip through different miRNAs, we tried to determine the absolute average copy numbers per single HeLa cell of the endogenous let-7a miRNA. We tailed total RNA of 0.02 μg/μl from HeLa, and captured them on a miRNA sensing chip. From more than three experiments, the average spot numbers of let-7a were counted as 17.3 ± 3.8, giving the concentrations of 6.2 pM (3,730,000 molecules/μl) when we used the calibration parameter of 2.8 spot/pM. Based on previous reports[28], we assumed that a single HeLa cell contained 20 pg of total RNA on average, and, therefore, 1 μl of 0.02 μg/μl total RNA solution contained total RNA from 1000 HeLa cells. From these considerations, we could estimate that a single HeLa cell contained 3730 ± 830 let-7a on average (Fig. 3f, left). To test whether Ago-FISH could monitor the dynamic change of miRNA expression, we quantified let-7a when it was overexpressed in various cells ("Method" section). We found that the let-7a number per single cell increased significantly (Fig. 3f and Supplementary Fig. 7) when let-7a was overexpressed. On the other hand, let-7a vanished (Supplementary Fig. 7) in Drosha knockout cells ("Method" section), confirming that Ago-FISH could reliably monitor the dynamic change of miRNA expression.

Finally, we demonstrated that Ago-FISH can be used to quantify miRNAs from various samples. We showed that human liver tissue expressed the increased copy numbers of both let-7a and let-7c compared to HeLa cell (Fig. 3g), which was consistent with the qRT-PCR results (Supplementary Fig. 8). We also demonstrated that Ago-FISH could quantify let-7a and let-7c miRNAs from MCF-7 exosomes (Fig. 3h). The quantities of the two miRNAs were significantly reduced in the exosome compared to MCF-7 cell (Fig. 3h).

In Ago-FISH, the average spot numbers, and thus the quantifications of miRNAs would be affected by the efficiencies of the RNA isolation, poly(A) tailing, and hybridization of miRNAs with biotinylated poly(T) DNA. We confirmed that the efficiencies of the poly(A) tailing and hybridization were almost perfect by comparing the spot numbers observed with biotinylated let-7a and poly(A)-tailed let-7a (Supplementary Fig. 9). We could not directly measure the RNA isolation efficiency, but observed the consistency of the efficiency of total RNA isolation by comparing the average let-7a spot numbers for three different batches of RNA isolation in the laboratory (17.3 ± 3.8, 18.0 ± 2.9, and 16.7 ± 3.9), and a commercial one (17.0 ± 2.0, Takara). These observations confirmed that our method of miRNA quantification was quite reliable.

## Discussion

In this work, we developed a highly reliable miRNA profiling technique, Ago-FISH, based on multicolor single-molecule imaging. Like the previously reported single-molecule miRNA detection technique, SiMREPS (Single-Molecule Recognition through Equilibrium Poisson Sampling)[17], Ago-FISH has the superior sensitivity, and the simplicity of workflow that does not require the enzymatic steps, such as amplification, reverse transcription, and ligation. Compared to SiMREPS, however, Ago-FISH has the merits of the increased miRNA detection speed, and the conservation of the high target specificity for the whole region of miRNAs. Furthermore, all miRNAs are captured in a single-miRNA sensing chip, and therefore, Ago-FISH can be converted to a high-throughput miRNA profiling technique when it is combined with microfluidics. The need to use purified proteins for miRNA detection, however, is a demerit of Ago-FISH. Compared to qRT-PCR, Ago-FISH is much more reliable due to reduced false positive rates, and not contaminated by pri- and pre-miRNAs (Supplementary Fig. 10). In this work, we demonstrated that miRNAs in the pM range can be reliably detected (Fig. 3e), but it should not be interpreted as the limit of Ago-FISH sensitivity. As already demonstrated in SiMREPS, the sensitivity of Ago-FISH can be improved to the fM range by increasing the miRNA capturing time[17]. The utilization of CMOS (complementary metal–oxide–semiconductor) camera instead of electron multiplying charge-coupled device (EM-CCD) will increase the imaging area, and thus the number of detected molecules per screen by an order of magnitude, and as a result the sensitivity of Ago-FISH can be improved further. Transient binding of DNA probes has been utilized for superresolution fluorescence microscopy, such as DNA-PAINT[29], but slow binding rates of DNA probes have hindered a wider application of the approach. Furthermore, endogenous miRNAs have never been imaged using DNA-PAINT. The accelerated target binding of $Tt$Ago-loaded DNA probes will allow the combination of Ago-FISH with superresolution fluorescence microscopy of endogenous miRNAs, which might provide valuable information on the spatial distribution of miRNAs in cells and tissues. In this work, Ago-FISH is used only to quantify miRNAs. The same technique, however, can be utilized to quantify other nucleic acids, such as mRNAs with accelerated binding rate, improved specificity, and not being limited by photobleaching.

## Methods

**Oligonucleotides preparation**. All synthetic oligonucleotides were purchased from Integrated DNA Technology (Coralville, IA). Amine-modified DNA probes were labeled with Cy3, Cy5, or Cy7 mono NHS-ester, which reacts with amine-reactive dyes (Cy3, Cy5, or Cy7 mono NHS-ester, GE Healthcare) by incubating 0.5 mM DNA oligos with 10 mM fluorophores in a reaction buffer (100 mM $Na_2BO_7$ pH 8.5) for 6 h. The excess dyes were removed by using ethanol precipitation. Labeled oligonucleotides were stored in 10 mM Tris-HCl (pH 8.0) with 50 mM NaCl. Information about oligo sequences is available in Supplementary Fig. 1.

**Purification of TtAgo**. TtAgo gene was cloned into a modified pET28a vector generating TtAgo protein with an N-terminal His-tag followed by TEV cleavage site. Ago was expressed in *Escherichia coli* BL21(DE3) cells with 0.5 mM IPTG at 18 °C. The cells expressing TtAgo protein were lysed by sonication in a lysis buffer containing 1 M NaCl, 20 mM Tris-HCl (pH 7.5), and 2 mM $MgCl_2$. The lysed cells were centrifuged at 40,000 × *g* for 1 h and the supernatant was incubated with Ni-NTA agarose resin (Qiagen) at 4 °C for 3 h. The Ni-NTA resin was then washed with the lysis buffer with 20 mM imidazole. TtAgo protein was eluted with a buffer containing 500 mM NaCl, 20 mM Tris-HCl (pH 7.5), 2 mM $MgCl_2$, and 100 mM imidazole, and the N-terminal His-tag was removed by TEV protease treatment. Thermostable TtAgo protein was further purified with heat treatment at 85 °C for 15 min. After the heat treatment, TtAgo proteins in soluble fraction were collected and further purified with Superdex S200 26/60 (GE Healthcare) size-exclusion column.

**Purification of endogenous total RNA HeLa (ATCC)**. HeLa cells were cultured in DMEM (Welgene) supplemented with 10% fetal bovine serum (Welgene), and total RNA was extracted using TRIzol reagent (Life Technologies). To overexpress let-7a, HeLa cells were transfected with pcDNA3-pri-let-7a-1 using Fugene HD Transfection Reagent (Promega), with 10 μg plasmid delivered per 100-mm dish. For mock samples, pcDNA3 was used for transfection. Cells were harvested 2 days after transfection. Commercial total RNAs from HeLa cells were purchased from Takara (636543).

*HCT116 (ATCC)*. HCT116 cells were maintained in McCoy's 5A (Welgene) supplemented with 10% fetal bovine serum (Welgene). To overexpress let-7a, HCT116 cells were transfected with pcDNA3-pri-let-7a-1 using Fugene HD Transfection Reagent (Promega), with 10 μg plasmid delivered per 100-mm dish. For mock samples, pcDNA3 was used for transfection. Cells were harvested 2 days after transfection. Drosha knockout procedure is described in the previous paper[30]. Total RNAs were isolated using TRIzol reagent (Life Technologies).

*HEK293E (ATCC)*. HEK293E cells were cultured in DMEM (Welgene) supplemented with 10% fetal bovine serum (Welgene). To overexpress let-7a, HEK293E cells were transfected with pcDNA3-pri-let-7a-1 using Fugene HD Transfection Reagent (Promega), with 10 μg plasmid delivered per 100-mm dish. For mock samples, pcDNA3 was used for transfection. Cells were harvested 2 days after transfection and total RNAs were isolated using TRIzol reagent (Life Technologies). Drosha knockout cells were generated by CRISPR/Cas9 technology.

*Liver tissue*. Total RNA from human liver tissue was purchased from Thermo Fisher (AM7960)

*MCF-7 (ATCC)*. MCF-7 cells were cultured at $6 \times 10^6$ per 150 mm dish in DMEM (HyClone) supplemented with 10% fetal bovine serum (Atlas) and 1% Antibiotic-Antimycotic (Gibco). Total RNA was extracted using the MiniBEST Universal RNA Exatraction Kit (Takara) following the manufacture's protocol. For exosomal total RNA purification, MCF-7 cells washed with DPBS (Welgene), and cultured in DMEM (HyClone) containing 1% Antibiotic-Antimycotic (Gibco) and 1% Glutamax (Gibco) for 2 days. After incubation for 2 days, exosomes were purified by serial centrifugation and filtration system. Briefly, the cell culture supernatants were centrifuged at 300 × *g* for 10 min, 2000 × *g* for 10 min, and 10,000 × *g* for 30 min to remove cell debris and microvesicles. A total of 2.4 L of supernatant was concentrated 35-fold with tangential flow filtration (TFF) using a 500 kDa mPES hollow fiber filter (D04-E500–05-N, Repligen) and KrosFlo KR2i TFF System (Repligen). A feed flow rate of 53 ml/min and a transmembrane pressure <1 bar were maintained during the concentration. The exosomes were pelleted by ultracentrifugation at 150,000 × *g* for 3 h in a 45 Ti rotor (Beckman Instruments). Exosomal total RNA was extracted using MiniBest Universal RNA Extraction Kit (Takara).

**3′-end poly(A) tailing of RNA**. Purified RNAs were heated at 75 °C for 3 min to avoid biases likely caused by the aggregations and secondary structures of RNAs[11,31]. Yeast poly(A) polymerase (Thermo Fisher) was used to add the poly(A) tail to RNA. Specifically, 0.1 μg/μl RNA was incubated at 37 °C for 1 h in a 20 mM Tris-HCl (pH 7.0) buffer with 20 μM EDTA, 0.2 mM DTT, 0.6 mM $MnCl_2$, 10% glycerol, 100 μg/ml acetylated BSA, 2.5 mM rATP, 40 U/μl yeast poly(A) polymerase (Thermo Fisher), and 0.5 U/μl RNase inhibitor (SUPERase.In™, Thermo

Fisher). The reaction was terminated by heating at 65 °C for 15 min. It was reported that the 3′-terminal nucleotide influences the poly(A) tail efficiency[11]. In our method, the capturing efficiency is not expected to vary significantly once the poly (A) tail is longer than 30 nt (the length of the complementary DNA use for capturing miRNAs). Furthermore, the influences of the 3′-terminal nucleotide on miRNA capturing efficiency could be effectively avoided by maintaining the poly (A) tailing reaction for 1 h (ref. [32]).

**Quantitative real-time PCR**. Purified total RNA was treated with DNase I (Takara). cDNAs were then synthesized using the Taqman miRNA Reverse Transcription Kit (Thermo Scientific) and subjected to qRT-PCR with the Taqman MicroRNA Assay (Applied Biosystems) on StepOnePlus Real-Time PCR System (Thermo Scientific).

**Single-molecule experiment**. Single-molecule experiments were performed in a total internal reflection fluorescence microscope. To reduce nonspecific binding of molecules, glass/quartz surface was coated with a mixture of PEG and biotin-PEG with 20:1 ratio. A detection chip was made between a quartz slide and a glass coverslip by using double-side sticky tape[33]. For experiments described in Fig. 3, 0.1 μg/μl poly(A)-tailed total RNA was incubated with 3 μM biotinylated poly(T) for 3 h. After the incubation, 0.1 μg/μl total RNA was diluted to 0.02 μg/μl using 10 mM Tris buffer (pH 8.0) with 50 mM NaCl, and then immobilized on surface by streptavidin–biotin interaction. For the experiments described in Figs. 1 and 2, biotinylated synthetic miRNAs were directly immobilized on a surface without the tailing and hybridization process. To assemble TtAgo and DNA probes, the three kinds of DNA probes (0.2 μM) was separately incubated with TtAgo (1 μM) for 30 min at 55 °C in an assembly buffer (10 mM Tris-HCl (pH 8.0) buffer with 100 mM NaCl, and 5 mM $MgCl_2$). TtAgo and DNA probes make a complex with a 1:1 stoichiometry, but excessive amount of TtAgo was used to reduce the amount of free DNA probes not loaded into TtAgo. For Ago-FISH experiments, the three kinds of TtAgo-loaded DNA probes were mixed, diluted to the final concentration of 2 nM in an imaging buffer (20 mM Tris-HCl (pH 8.0) buffer with 135 mM KCl, 0.5% formamide, 120 mM UREA, 1 mM $Mg^{2+}$, and oxygen scavenger system: 5 mM protocatechuic acid, 500 nM protocatechuate 3,4-dioxygenase, and saturated Trolox (25 mg/50 ml)), and injected into the detection chamber. Single-molecule fluorescence imaging experiments were started right after the injection of DNA probes at 30 °C. Cy3, Cy5, and Cy7 were excited by a 532-nm laser (Compass215M, Coherent), a 640-nm laser (Cube640–100C, Coherent), and a 730-nm laser (PhoxX® 730, Omicron), respectively. The laser intensities were adjusted to obtain similar signal intensities of the three dyes. For alternative laser excitation (ALEX) used for data collection, three mechanical shutters (LS-3, Uniblitz, Rochester, NY) were alternatively switched every 0.5 s synchronously with the frame rate of the camera, giving the overall time resolution of imaging as 1.5 s. Fluorescence signals of Cy3, Cy5, and Cy7 were collected through a water-immersion objective (UPlanSApo 60×, Olympus), separated by using two dichroic mirrors (635dcxr and 740dcxr, Chroma), and imaged on an EM-CCD camera (Ixon DV897, Andor). Data were collected by using home-built program written in Visual C++ (Microsoft), and analyzed by using IDL (7.0, ITT), MATLAB (R2010a, The MathWorks), and Origin (8.0, OriginLab).

**Reporting summary**. Further information on research design is available in the Nature Research Reporting Summary linked to this article.

## Data availability

A reporting summary for this article is available as a Supplementary Information file. All other relevant data are available from the corresponding authors upon reasonable request. Source data are provided with this paper.

## Code availability

The custom scripts are available from the corresponding authors upon reasonable request.

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

## Acknowledgements

This work was supported by grants from the National Research Foundation of Korea to S.H. (NRF-2019R1A2C2005209) and to J.S. (NRF-2020R1A2B5B03001517), the KIST Institutional Program to C.J. and by the Institute for Basic Science to V.N.K. (IBS-R008-D1).

## Author contributions

S.H. supervised the research. S.S., H.U., and M.S. performed single-molecule experiments. Y.J. and S.S. performed total RNA purification and qRT-PCR experiments. J.G. and C.J. contributed to exosomal RNA purification. J.S. provided Argonaute protein. All contributed to writing of the paper.

## Competing interests

The authors declare no competing interests.
