## [Peer Review File · Nature Communications]

Reviewers' Comments:

Reviewer #1:

Remarks to the Author:

In this manuscript, Dr. Hohng and colleagues describe a new method for miRNA detection, which they call miRQ-ALP. miRQ-ALP is a method that relies on a hybridization of three short oligonucleotides coupled to different fluorescent dyes to three different regions on a target miRNA. The method works on RNA extracted from cells after the RNA is coupled to a glass surface through biotin-streptavidin interaction. In order to speed up the detection, fluorescently-coupled oligonucleotides are pre-loaded to the *Thermus thermophilus* Argonaute protein and the complex is then used for hybridization. The advantages of the method seem to include the speed of detection, sensitivity, specificity and the ability to profile the entire length of miRNAs. The latter is especially relevant in cases where point mutations might be present in different parts of miRNAs of interest. As such, the method seems pretty straightforward and relatively easy. The data on specificity as well as on the kinetics seem convincing. The method addresses an important problem in the area of FISH-based detection methods, given that detection of miRNAs is still a challenge. It also promises to find many relevant applications given the importance of the miRNAs research. However, the description of the method is rather limited, the entire manuscript is very short and often lacks important details. The description should walk the reader through the design of the method step by step to avoid the need to reread the text multiple times in order to understand it. Also, detailed information about the quantitation of the signals is missing. It is difficult to gain an insight into how sensitive the method truly is and it is relatively difficult to judge the overall benefits of this method over other available techniques. Therefore, the manuscript needs to become much richer in detail in order for a reader to grasp the true potential of this method.

Here I list my major comments/critique:

The paragraph on the comparison of the new method with existing ones is poorly structured. It should be done in a more thorough way where the main parameters like sensitivity, specificity, resolution (single-cell?), ease of performing the assay and cost-effectiveness is compared for all the methods (in addition to method-specific characteristics). Some statements in this section are vague, such as the sequencing being "poor in precision". Also the statement about biases of the RT-PCR is vague (which biases?).

I have a difficult time understanding how a typical experiment of the miRQ-ALP looks like as the description of the method is very obscure. For example:

- why is the PS oligo shorter than others?
- why the different oligos (PS, PM and PT) have a different sequence to which dyes are conjugated (the part other than the complementary part)
- in a typical experiment, how many signals will show only 1 out of 3 colors, or only 2 out of 3 colors?
- How many spots per experiment are analyzed? (I would like to see some statistics on all the spots analyzed per experiment and some characterization of reproducibility)
- What is the typical imaging mode? What are the excitation times and the frequency of excitation? Can differences of the illumination parameters between dyes affect the result? Can they explain why the Cy3 signal has much faster kinetics (despite the oligo having a longer complementary part than the Cy7) than the other two?
- How long is the typical imaging time? How many time does one need to see a certain signal (color) (with which kinetics) to consider a spot positive? For example, in the experiment shown in Fig. 3 c, in the lower panel where the signal from let-7c is shown, there are only 3 instances of the Cy7 signal. Is that enough? If there were one only, should one consider it as positive or negative? I understand that in this specific case, given that one needs to see 3 colors the let-7c signal (in the first half of the imaging) would not be scored as positive. However, this panel, given that the Cy7 signal should be normally present, stimulated my questions about what should one consider as a prerequisite to call a signal positive. In one part there is the threshold of 5% duty cycle specified

but I could not follow this reasoning. These considerations should be discussed and clear guidelines for others provided.

The part about benchmarking (lines 89-94), especially the part about the choice of the 5% duty cycle, is not understandable.

It is not clear whether the entire RNA pool is attached to the glass surface for the assay. If so, could the Authors explore the possibility of first pre-selecting only for short RNAs after removing the ribosomal RNA? Could the method benefit from such a pre-treatment?

Considerations on the sensitivity of the method are missing. How abundant does an miRNA need to be in order for this method to detect it. The Authors should perform some depletion experiments to demonstrate that the method is sensitive enough to detect such changes in the endogenous RNA preparation.

Is the biotinylation necessary? Have the Authors tried the RNA spotting method described in Semrau et al. Cell Reports 2014? If not, I suggest to try it instead in case it makes the assay more straightforward.

The need to use purified protein makes the assay rather cumbersome. Can the Authors discuss this aspect in the Discussion?

When the Authors provide the estimate of the numbers of let-7a and let-7c in HeLa cells, per which spotted volume were the numbers they provided (17.3 +/- 3.8 as spot numbers) for? This part is not clear.

The use of 75C heating for 3 minutes is not justified in the miRNA preparation step. Is that truly necessary? RNA is known for not tolerating heating well. Is this step used to destroy secondary structure? If this is the case, can the Authors explain why they think their method does not detect the pre-miRNA?

What is the part explained in lines 244-248 for? There should be an explanation for what the incubation the Authors describe (with DTT, glycerol, BSA etc.) for.

The statement that the let-7a is much more abundant than the let-7c, based on the experiment presented in Figure 3e seems exaggerated. I do see a difference but I do not know where the "much more" description comes from.

The Authors write about the possible application of their method to super-resolution microscopy (I assume for imaging miRNA inside cells? This is missing.). How could that be achieved if the method relies on a bulky complex of oligos with Argonaute? Do the Authors envision microinjections as a possible adaptation of this method to single-cells? Could the Authors explain this better?

Minor comments:

The abstract should contain an information about the format of the assay, whether the method is performed in intact cells or on a cell-free RNA extract. In its current form it is confusing.

The detection chip description is rather poor: "A detection chip was made between a quartz slide and a glass coverslip by using double-side sticky tape." Perhaps the Authors could draw a scheme of it to facilitate other groups reproducing their results.

I suggest changing the name of the technique to one that contains the name "FISH" in it and is easier to pronounce.

In the introduction, when the Authors introduce the relevance of miRNA and write about high abundance of miRNAs in blood, they should provide information about the fact that miRNAs are of higher abundance in cancer patients in comparison to healthy individuals. On its own high abundance does not warrant good therapy target. Slight rephrasing is needed to make the statement more accurate and up to the point.

I would suggest rephrasing the sentence that starts in line 44.
The part "but they have their own merits and demerits" does not flow well.

The Authors should avoid vague statements such as "extremely slow process" (line 65) and instead provide approximate numbers.

The Authors should introduce the rationale behind using the TtAgo.

Before the Authors use the term "duty cycle" they should first introduce it.

The sentence that starts in line 97 should be corrected (the "polymerases adenosine tails" part).

In Fig. 3a there seems to be an error (Poly (X) instead of Poly (A)).

Reviewer #2:

Remarks to the Author:

In the study by Shin and co-workers, the authors describe a method for single-molecule miRNA detection that is a variation of that reported by Johnson-Buck et al. in 2015. In the current study, miRNA probes were loaded into recombinant *Thermus thermophilus* Argonaute (TtAgo) and incubated with target RNAs immobilized on a polymer-coated quartz surface. Binding of miRNA probes were monitored using a multi-color single-molecule fluorescence microscope. The authors show the TtAgo-loaded miRNA probes had an increase binding rate compared to bare probes, were able to distinguish single nucleotide mutations in target miRNAs, and were able to quantify endogenous miRNA levels in total RNA from HeLa cells. Overall this is an interesting study featuring an innovative miRNA profiling technique, miRQ-ALP, that is sensitive, specific and does not require enzymatic amplification. However, additional information about experiments described in this study and the use of this technique should be provided so that this technique can be more widely useful.

Specific comment/concerns:

1. Fig. 1. Please clarify the stoichiometry of DNA probe loaded into TtAgo. Is the ratio of probe to TtAgo the same for all probes? Please clarify whether the experiment in Fig. 1d involves incubation of the three probes with target simultaneously or separately. Please provide rationale for design of probes binding to seed, mid and tail segments of target miRNA and whether GC content and probe length was important for probe design. Please provide data for control experiment conditions – fluorescence intensity time traces without target or in the presence of probe competitor.

2. Fig. 2b. It is not clear why intensity time traces for a probe look different for mutant targets that do not have a mutation in probe's complementary sequence. For example, the time traces of Cy7 were different for let-7a_m2 and let-7a_m3. The probe labeling method and experimental setup for Fig. 2e described in Supplemental Tables 1 and 2 is very confusing.

3. Fig. 3. No information is provided on the efficiency of the endogenous immobilization process, which required RNA isolation, addition of polyA tails, and hybridization with biotinylated polyT DNA. Please show control experiment findings for Fig. 3b. It would also be important to show that this technique can capture dynamic changes in miRNA expression, such as when cells are stimulated.

How would RNA isolation and immobilization be normalized for comparison between different experimental conditions?

4. In Fig. 3, it is not surprising that the slopes for the experiments in 3d are similar to those in 3e, since the levels of spike-in miRNA are much greater than endogenous miRNA. What are the slopes when <1 pM of synthetic miRNA is spiked-in, which would be comparable to endogenous levels. It is not clear that the calibration parameter of 2.8 spot/pM for quantification of miRNA levels in 3f is accurate based on curves in 3e where the lowest concentration of spike-in was 25 pM.

5. Please clarify the amount of synthetic and spike-in miRNA in Supplementary Fig. 1. Details of the qRT-PCR experiment in Supplementary Fig. 2 are missing. For example, were Ct values normalized to small RNA? Please provide additional information about data presented in Supplementary Fig. 3. Why do the fluorescent miRQ-ALP probes not bind to mature miRNA sequence in the pre-miRNA? Is this phenomenon dependent on incubation temperature?

4. Profiling of miRNAs in plasma or serum would be an important application of the miRQ-ALP technique. Is this method able to detect extracellular miRNAs in plasma or serum?

Reviewer #3:

Remarks to the Author:

The article by Shin et al, describes a single-molecule miRNA detection platform that uses Ago-bound fluorescent probes. This is accomplished by visualizing Ago-probe association with immobilized target miRNAs using super resolution microscopy. The technology is specific, being able to distinguish single base pair differences, and is less fraught with artifacts when compared to qPCR methods. The authors demonstrate the specificity and sensitivity using synthetic oligos and purified total RNA. They go on to provide an exact calculation of let-7 molecules in a HeLa cell.

The technology appears to be a reliable platform, and the experiments demonstrating its performance are satisfying. Overall though this is more of an evolutionary development rather than revolutionary. Demonstrating the application described at the end of manuscript (in vivo, single molecule miRNA imaging with super resolution microscopy) would likely garner more interest in the article. The authors also "over-sell" the platform in a couple ways that should be addressed. The data in the manuscript are well-presented, however, I have several recommendations for the authors.

1) The most significant change I recommend is providing additional panels for figure 3b that shows let-7c spots. It would be compelling to see distinct spots correspond to let-7a and let-7c for the tail probe upon changing the probe set.

2) The authors point out that qPCR suffers from enzyme-introduced artifacts. The platform presented by the article requires polyA polymerase modification of target RNAs. There are biases associated with this enzyme as well. The authors should qualify these statements, or demonstrate that there is no bias associated with polyA polymerase.

3) It is unclear why the authors focus exclusively on microRNAs. It would seem that any nucleic acid sequence could be targeted by the Ago probes. TtAgo is a DNA binding Ago and does not target RNAs in vivo. Seems that binding could occur on any immobilized nucleic acid. This opens many applications that would benefit from the enhanced, prolonged, specific binding of the Ago probes.

4) The authors initially describe the platform as having implications for the clinic. While the target binding aspect of the platform can certainly be miniaturized to a "chip", the super resolution

microscopy needed to read the chip is not going to be a "rapid" option due to equipment being unavailable outside the most well-equipped medical center.

Response to the Comments of Referee #1:

Comments for the Authors

In this manuscript, Dr. Hohng and colleagues describe a new method for miRNA detection, which they call miRQ-ALP. miRQ-ALP is a method that relies on a hybridization of three short oligonucleotides coupled to different fluorescent dyes to three different regions on a target miRNA. The method works on RNA extracted from cells after the RNA is coupled to a glass surface through biotin-streptavidin interaction. In order to speed up the detection, fluorescently-coupled oligonucleotides are pre-loaded to the Thermus thermophiles Argonaute protein and the complex is then used for hybridization. The advantages of the method seem to include the speed of detection, sensitivity, specificity and the ability to profile the entire length of miRNAs. The latter is especially relevant in cases where point mutations might be present in different parts of miRNAs of interest. As such, the method seems pretty straightforward and relatively easy. The data on specificity as well as on the kinetics seem convincing. The method addresses an important problem in the area of FISH-based detection methods, given that detection of miRNAs is still a challenge. It also promises to find many relevant applications given the importance of the miRNAs research. However, the description of the method is rather limited, the entire manuscript is very short and often lacks important details. The description should walk the reader through the design of the method step by step to avoid the need to reread the text multiple times in order to understand it. Also, detailed information about the quantitation of the signals is missing. It is difficult to gain an insight into how sensitive the method truly is and it is relatively difficult to judge the overall benefits of this method over other available techniques. Therefore, the manuscript needs to become much richer in detail in order for a reader to grasp the true potential of this method.

We would like to thank the Reviewer for the favorable evaluation of our paper, and we tried to address all the concerns raised by the Reviewer as shown below.

Here I list my major comments/critique:

The paragraph on the comparison of the new method with existing ones is poorly structured. It should be done in a more thorough way where the main parameters like sensitivity, specificity, resolution (single-cell?), ease of performing the assay and cost-effectiveness is compared for all the methods (in addition to method-specific characteristics). Some statements in this section are vague, such as the sequencing being "poor in precision". Also the statement about biases of the RT-PCR is vague (which biases?).

Following the suggestion, we revised our manuscript as shown below. We could not compare all commercialized miRNA detection method by ourselves. Instead, we referred the references that previously addressed the question.

"A number of miRNA detection techniques are currently commercialized, but they have their own merits and demerits. For example, microarray analysis can monitor large number of target miRNAs in a cost-efficient way, but has a poor specificity and limited dynamic range. RNA sequencing is a technique with a high throughput, but rather expensive and poor in precision. Quantitative real-time PCR (qRT-PCR) is relatively precise, but prone to bias and can be easily contaminated by pri- and pre-miRNAs."

→ "A number of miRNA detection techniques are currently commercialized, but they have their own limitations¹⁰⁻¹⁴. In a conventional microarray analysis, extracted miRNAs are poly(A)-tailed, ligated with a biotinylated DNA strand, and captured on an array of spots of a microarray plate, each of which are coated with specific DNA strands complementary to the target miRNA. Binding of miRNAs on the spot is detected by adding fluorescently labeled streptavidin. Using this technique, we

can monitor a large number of target miRNAs in low cost, but the specificity and dynamic range of microarray analysis are limited because a RNA-DNA duplex of 20 base pairs is quite stable, and as a result, single base pair difference cannot be detected. In a conventional miRNA sequencing, extracted miRNAs are ligated with RNA adapters on both the 3'- and 5'-sides, converted into DNA strands by reverse transcriptase, and amplified using PCR (Polymerase Chain Reaction). After each enzymatic step, the reaction products should be purified using PAGE (Polyacrylamide gel electrophoresis), and the final products are sequenced using a sequencing machine. The RNA sequencing technique is suitable for high throughput miRNA profiling, and can discover new miRNAs, but rather expensive and its precision of quantitation is poor. Furthermore, uneven enzymatic reactions cause bias in miRNA detection. Recently the bias problem of RNA sequencing was improved by adopting random adapters and crowding reaction conditions, but not completely solved¹⁵. In a conventional quantitative real-time PCR (qRT-PCR), extracted miRNAs are converted into DNA using reverse transcription, and progress of quantitative PCR is monitored using a special fluorescent probe. qRT-PCR is known to be relatively precise, but prone to bias and can be easily contaminated by pri- and pre-miRNAs^{13,16}.

I have a difficult time understanding how a typical experiment of the miRQ-ALP looks like as the description of the method is very obscure. For example:

- why is the PS oligo shorter than others?

To address the comment, we made revisions through the manuscript. We also added the following sentence to the manuscript.

“When we design DNA probes, the lengths of the complementary part of the DNA probes were optimized so that they have comparable binding/dissociation kinetics on target miRNAs by compensating the varying GC contents of the seed, mid, and tail regions.”

- why the different oligos (PS, PM and PT) have a different sequence to which dyes are conjugated (the part other than the complementary part)?

To address the question, we added the following sentence to the manuscript.

“The sequence of non-complementary part of DNA probes is not critical but should be selected for DNA probes not to form secondary structures. The overall length of DNA probes was maintained at 21 nt.”

- in a typical experiment, how many signals will show only 1 out of 3 colors, or only 2 out of 3 colors?

To address the question, we revised the manuscript as follows.

“Quantitatively, the average duty cycle of the mutated region-targeting probes decreased below 0.05 compared to the original values of 0.19, 0.17, and 0.23 of let-7a_PS, let-7a_PM, and let-7a_PT, respectively (Fig. 2c).”

→

“For a quantitative analysis, we selected all time traces which showed the binding of any of the three DNA probes during observation, and obtained the average duty cycle (the fraction of observation time in which a probe is binding to the target miRNA) of each probe. Fig. 2c shows that the average duty cycle of the mutated region-targeting probes decreased significantly compared to the original values of 19%, 17%, and 23% of let-7a_PS, let-7a_PM, and let-7a_PT, respectively (Fig. 2c). In the case of wild type let-7a, most of traces (98.2%) showed the binding of the three DNA probes except the minor portion of traces which showed either one-probe binding (0.5 %), or two-probe binding (1.3 %), demonstrating that Ago-FISH can detect target miRNAs with a fidelity.”

- How many spots per experiment are analyzed? (I would like to see some statistics on all the spots analyzed per experiment and some characterization of reproducibility)

To obtain data points, we conducted more than three independent experiments and at least 300 molecules in total were analyzed. The information is added to the captions of Fig. 1-2.

- What is the typical imaging mode? What are the excitation times and the frequency of excitation? Can differences of the illumination parameters between dyes affect the result? Can they explain why the Cy3 signal has much faster kinetics (despite the oligo having a longer complementary part than the Cy7) than the other two?

To better address the questions, we revised the Method section as shown below. We do not clearly understand why the Cy3 signal has faster kinetics than the Cy7 signal even though it has a longer complementary part, but guess that a possible secondary structure of the target miRNA formed in the mid region (please see below) affects the binding/dissociation kinetics.

“Cy3, Cy5, and Cy7 were excited by a 532-nm laser (Compass215M, Coherent), by a 640-nm laser (Cube640-100C, Coherent), and by a 730-nm laser (PhoxX® 730, Omicron), respectively. To switch three lasers for alternative laser excitation (ALEX), mechanical shutters (LS-3, Uniblitz, Rochester, NY) were used.”

→

“Cy3, Cy5, and Cy7 were excited by a 532-nm laser (Compass215M, Coherent), a 640-nm laser (Cube640-100C, Coherent), and a 730-nm laser (PhoxX® 730, Omicron), respectively. **The laser intensities were adjusted to obtain similar signal intensities of the three dyes. For alternative laser excitation (ALEX) used for data collection, three mechanical shutters (LS-3, Uniblitz, Rochester, NY) were alternatively switched every 0.5 s synchronously with the frame rate of the camera, giving the overall time resolution of imaging as 1.5 s.**”

- How long is the typical imaging time?

Imaging time varied from 10 to 25 minutes, and the typical one was 20 minutes. However, this should not be considered as the limit of Ago-FISH imaging speed because we did not try to accelerate the imaging speed in this work.

How many time does one need to see a certain signal (color) (with which kinetics) to consider a spot positive? For example, in the experiment shown in Fig. 3 c, in the lower panel where the signal from

let-7c is shown, there are only 3 instances of the Cy7 signal. Is that enough? If there were one only, should one consider it as positive or negative? I understand that in this specific case, given that one needs to see 3 colors the let-7c signal (in the first half of the imaging) would not be scored as positive. However, this panel, given that the Cy7 signal should be normally present, stimulated my questions about what should one consider as a prerequisite to call a signal positive. In one part there is the threshold of 5% duty cycle specified but I could not follow this reasoning. These considerations should be discussed and clear guidelines for others provided. The part about benchmarking (lines 89-94), especially the part about the choice of the 5% duty cycle, is not understandable.

To clarify the meaning of the duty cycle, we added its definition to the manuscript as shown below.

“For a quantitative analysis, we selected all time traces which showed the binding of any of the three DNA probes during observation, and obtained the average duty cycle (the fraction of observation time in which a probe is binding to the target miRNA) of each probe.”

To clarify the criteria by which the true positive and the false negative are determined, we revised the manuscript as follows. As you can see in the revised manuscript, we tested two different approaches for the estimation of false negative rates using 5% duty cycle threshold, and the both approaches revealed that Ago-FISH can provide lower false positive rates than qRT-PCR. We also showed that the false positive rates of Ago-FISH becomes almost zero if we use 10% duty cycle threshold. Please notice that we added to the revision a new data showing that Ago-FISH can distinguish the same miRNA with different 3'-tails with high specificity.

“To benchmark miRQ-ALP against commercialized qRT-PCR techniques, we chose 5% duty cycle of probe binding as a threshold to identify as a proper target miRNA, and quantified the false positive rates of let-7 family miRNAs (Fig. 2d) using DNA probes optimized for their detection (Supplementary Table 1-2). The false positive rates are significantly reduced in miRQ-ALP (Fig. 2e) compared to those of commercial qRT-PCR techniques (Supplementary Table 3)¹⁸.”

→

“To benchmark Ago-FISH against commercialized qRT-PCR techniques, we chose 5% duty cycle of probe binding as a threshold (i.e. we consider a molecule as positive when the duty cycles of all of the three DNA probes are larger than 5 %) to identify as a proper target miRNA in Ago-FISH, and quantified the true and false positive rates of let-7 family miRNAs (Fig. 2d) using DNA probes optimized for their detection (Supplementary Fig. 1). The false positive rates are significantly reduced in Ago-FISH (Fig. 2e) compared to those of commercial qRT-PCR techniques (Supplementary Fig. 2)¹³. A different approach to estimate a false positive rate is to compare the average spot numbers of let-7 family miRNAs (relative detection), in which the true positive rate is automatically assumed as 100% as in qRT-PCR. This approach also revealed improved false positive rates of Ago-FISH compared to those of qRT-PCR (Supplementary Fig. 3).”

During maturation, the 3'-end of miRNA can be modified by terminal nucleotidyl transferases, and this RNA tailing affects downstream processing and miRNA stability²⁵⁻²⁷. We tested whether Ago-FISH can reliably discriminate mono- and bi-uridylation of miRNAs using let-7a miRNAs with different tail regions (Fig. 2f) and DNA probes optimized for their detection (Supplementary Fig. 1). Fig. 2g shows that Ago-FISH can distinguish mono- and bi-uridylated miRNAs with high specificity. In Fig. 2e and Fig. 2g, we arbitrarily chose 5% as the duty cycle threshold. When a better specificity is required, however, the higher threshold value can be used by sacrificing the detection efficiency (defined as the ratio of the true positive to the sum of the true positive and the false negative) (Supplementary Fig. 3-5).”

It is not clear whether the entire RNA pool is attached to the glass surface for the assay. If so, could the Authors explore the possibility of first pre-selecting only for short RNAs after removing the ribosomal RNA? Could the method benefit from such a pre-treatment?

We could detect miRNAs after purifying short RNAs, but found that the purification yield varies for each trial, and miRNAs were lost during the purification. Without a clear advantage of the purification step, it was not included into the protocol of Ago-FISH. To clarify the point, we added the following sentences to the manuscript.

“When we first prepared endogenous RNAs, we tried to additionally purify short RNAs from total RNAs. However, the addition of the short RNA purification step did not produce any improvement in miRNA detection, but only problems (the purification yield varied from trial to trial, and miRNAs were lost during the additional purification step), and therefore was not used in this work.”

Considerations on the sensitivity of the method are missing. How abundant does a miRNA need to be in order for this method to detect it. The Authors should perform some depletion experiments to demonstrate that the method is sensitive enough to detect such changes in the endogenous RNA preparation.

As shown in Fig. 3e, miRNAs in pM range can be reliably detected in the current setting. However, we expect that the sensitivity of Ago-FISH can be improved to fM range by increasing the miRNA capturing time as already demonstrated by Nils Walter’s group. The usage of CMOS camera instead of EM-CCD camera will increase the imaging area, and thus the number of detected molecules per screen by an order of magnitude, and as a result the sensitivity of Ago-FISH can be improved as such an amount. To clarify the point we added the following sentences to the manuscript.

“In this work, we demonstrated that miRNAs in the pM range can be reliably detected (Fig. 3e), but it should not be interpreted as the limit of Ago-FISH sensitivity. As already demonstrated in SiMREPS, the sensitivity of Ago-FISH can be improved to the fM range by increasing the miRNA capturing time¹⁷. The utilization of CMOS (Complementary Metal–Oxide–Semiconductor) camera instead of EM-CCD (Electron Multiplying-Charge Coupled Device) will increase the imaging area, and thus the number of detected molecules per screen by an order of magnitude, and as a result the sensitivity of Ago-FISH can be improved further.”

Is the biotinylation necessary? Have the Authors tried the RNA spotting method described in Semrau et al. Cell Reports 2014? If not, I suggest to try it instead in case it makes the assay more straightforward.

Thank you for the suggestion. In this work, we have not tried the RNA spotting method, but will adopt the method in future works if it works better than the current protocol.

The need to use purified protein makes the assay rather cumbersome. Can the Authors discuss this aspect in the Discussion?

Following the suggestion, we added the following sentence to the manuscript.

“The need to use purified proteins for miRNA detection, however, is a demerit of Ago-FISH.”

When the Authors provide the estimate of the numbers of let-7a and let-7c in HeLa cells, per which spotted volume were the numbers they provided (17.3 +/- 3.8 as spot numbers) for? This part is not clear.

The given values are the average spot numbers per imaging area. To clarify the point, we revised the manuscript as follows.

“The number of spots identified as let-7a and let-7c increased linearly with their concentrations during the immobilization step, giving the similar slopes of 2.7 spot/pM for let-7a, and 2.8 spot/pM for let-7c (Fig. 3d).”

→

“The **average** number of spots **per imaging area (20 μm x 60 μm)** identified as let-7a and let-7c **using 5% duty cycle threshold** increased linearly with their concentrations during the immobilization step, giving the similar slopes of **2.8** spot/pM for let-7a, and **2.7** spot/pM for let-7c (Fig. 3d).”

The use of 75°C heating for 3 minutes is not justified in the miRNA preparation step. Is that truly necessary? RNA is known for not tolerating heating well. Is this step used to destroy secondary structure?

Heating at 75°C prior to RNA tailing was performed to destroy aggregations and secondary structures of RNA. We found this step is necessary for unbiased poly (A) tailing. To clarify the point, we revised the manuscript as follows.

“Total RNA samples were heated at 75°C for 3 minutes and immediately placed on ice. Then 0.1 μg/μl total RNA were incubated at 37°C for 1 hour with 20 mM Tris-HCl (pH 7.0), 20 μM EDTA, 0.2 mM DTT, 0.6 mM MnCl₂, 10% glycerol, 100 μg/ml acetylated BSA, 2.5 mM rATP, Yeast Poly(A) Polymerase (Thermo Fisher) 40 U/μl, and RNase inhibitor (SUPERase.In™, Thermo Fisher) 0.5 U/μl. The reaction was terminated by heating at 65°C for 15 minutes.”

→

“**Purified RNAs were heated at 75°C for 3 minutes to avoid biases likely caused by the aggregations and secondary structures of RNAs^{11,31}. Yeast poly(A) polymerase (Thermo Fisher) was used to add the poly(A) tail to RNA. Specifically, 0.1 μg/μl RNA was incubated at 37 °C for 1 hour in a 20 mM Tris-HCl (pH 7.0) buffer with 20 μM EDTA, 0.2 mM DTT, 0.6 mM MnCl₂, 10% glycerol, 100 μg/ml acetylated BSA, 2.5 mM rATP, 40 U/μl yeast poly(A) polymerase (Thermo Fisher), and 0.5 U/μl RNase inhibitor (SUPERase.In™, Thermo Fisher). The reaction was terminated by heating at 65°C for 15 minutes.**”

If this is the case, can the Authors explain why they think their method does not detect the pre-miRNA?

Ago-FISH data are not contaminated by pre-miRNAs because miRNA detection is conducted at 30°C where secondary structure of pre-miRNA is stable. To clarify the point, we revised the caption of Supplementary Fig. 10 as follows.

“None of the pre-let-7a was detected.”

→

“**miRNA detection experiment was conducted at 30 °C where secondary structure of pre-miRNA is stable. Therefore, DNA probes cannot access the target region of pre-miRNA, and none of the pre-let-7a was detected.**”

What is the part explained in lines 244-248 for? There should be an explanation for what the incubation the Authors describe (with DTT, glycerol, BSA etc.) for.

To address the comment, the manuscript is revised as follows.

“Then 0.1 μg/μl total RNA were incubated at 37°C for 1 hour with 20 mM Tris-HCl (pH 7.0), 20 μM EDTA, 0.2 mM DTT, 0.6 mM MnCl₂, 10% glycerol, 100 μg/ml acetylated BSA, 2.5 mM rATP, Yeast Poly(A) Polymerase (Thermo Fisher) 40 U/μl, and RNase inhibitor (SUPERase.In™, Thermo Fisher)

0.5 U/ μ l.”

→

“Yeast poly(A) polymerase (Thermo Fisher) was used to add the poly(A) tail to RNA. Specifically, 0.1 μ g/ μ l RNA was incubated at 37 °C for 1 hour in a 20 mM Tris-HCl (pH 7.0) buffer with 20 μ M EDTA, 0.2 mM DTT, 0.6 mM MnCl₂, 10% glycerol, 100 μ g/ml acetylated BSA, 2.5 mM rATP, 40 U/ μ l yeast poly(A) polymerase (Thermo Fisher), and 0.5 U/ μ l RNase inhibitor (SUPERase.InTm, Thermo Fisher).”

The statement that the let-7a is much more abundant than the let-7c, based on the experiment presented in Figure 3e seems exaggerated. I do see a difference but I do not know where the “much more” description comes from.

To address the comment, we performed the experiments at lower concentrations of injected miRNAs. Please see the revised Fig. 3e. The manuscript is also revised as follows.

“The y-intercepts of the let-7a data, however, was clearly larger than that of let-7c data (Fig. 3e), indicating let-7a is much more abundant than let-7c in HeLa cell, which is consistent with the RNA sequencing results (data not shown).”

→

“The y-intercepts of the let-7a data, however, was **13 times** larger than that of let-7c data (**17.5 vs. 1.3**, Fig. 3e), indicating let-7a is more abundant than let-7c in HeLa cell, which is consistent with the RNA sequencing results (data not shown).”

The Authors write about the possible application of their method to super-resolution microscopy (I assume for imaging miRNA inside cells? This is missing.). How could that be achieved if the method relies on a bulky complex of oligos with Argonaute? Do the Authors envision microinjections as a possible adaptation of this method to single-cells? Could the Authors explain this better?

Following the suggestion, we revised the manuscript as follows.

“The accelerated target binding of *Tt*Ago-loaded DNA probes will allow the combination of miRQ-ALP with superresolution fluorescence microscopy^{12,20}, which will provide the valuable information on the spatial distribution of miRNAs in cells and tissues.”

→

“Transient binding of DNA probes has been utilized for superresolution fluorescence microscopy such as DNA-PAINT²⁹, but slow binding rates of DNA probes have hindered a wider application of the approach. Furthermore, endogenous miRNAs have never been imaged using DNA-PAINT. The accelerated target binding of *Tt*Ago-loaded DNA probes will allow the combination of Ago-FISH with superresolution fluorescence microscopy of endogenous miRNAs, which will provide the valuable information on the spatial distribution of miRNAs in cells and tissues.”

Minor comments:

The abstract should contain information about the format of the assay, whether the method is performed in intact cells or on a cell-free RNA extract. In its current form it is confusing.

Following the suggestion, we revised the title and abstract as follows.

“Quantification of endogenous miRNAs with high sensitivity, specificity, and reliability”

→

“Quantification of **purified** endogenous miRNAs with high sensitivity, specificity, and reliability”

“Here we report an amplification-free multi-color single-molecule imaging technique that can profile endogenous miRNAs with high sensitivity, specificity, and reliability.”

→

“Here we report an amplification-free multi-color single-molecule imaging technique that can profile **purified endogenous miRNAs with high sensitivity, specificity, and reliability.”**

The detection chip description is rather poor: “A detection chip was made between a quartz slide and a glass coverslip by using double-side sticky tape.” Perhaps the Authors could draw a scheme of it to facilitate other groups reproducing their results.

The protocol of the miRNA detection chip construction is a standard one widely used in the single-molecule research field. Instead of repeating the details of the protocol, we added a reference to the manuscript.

I suggest changing the name of the technique to one that contains the name “FISH” in it and is easier to pronounce.

In general, FISH is performed in fixed cells, and should provide the positional information of mRNAs inside cells. In that respect, FISH is not an adequate name for the current version of our technique. However, our technique has a potential to be used to image and quantify miRNAs in fixed cells, and the Reviewer’s comment seems relevant. Following the suggestion, we renamed our technique as Ago-FISH (Argonaute-based FISH).

In the introduction, when the Authors introduce the relevance of miRNA and write about high abundance of miRNAs in blood, they should provide information about the fact that miRNAs are of higher abundance in cancer patients in comparison to healthy individuals. On its own high abundance does not warrant good therapy target. Slight rephrasing is needed to make the statement more accurate and up to the point.

Following the suggestion, the manuscript is revised as follows.

“For instance, a large number of miRNAs are found to be dysregulated in a broad spectrum of cancers in a disease-specific fashion⁴, and specific miRNAs circulating in the blood of numerous cancer patients were found to be highly abundant, and remarkably stable⁵, making miRNAs ideal tumor markers for early detection.”

→

“For instance, a large number of miRNAs are found to be dysregulated in a broad spectrum of cancers in a disease-specific fashion⁴, and specific miRNAs circulating in the blood of numerous cancer patients were found to be highly abundant, and remarkably stable **compared to those in the blood of normal people**⁵, making miRNAs ideal tumor markers for early detection.”

I would suggest rephrasing the sentence that starts in line 44. The part “but they have their own merits and demerits” does not flow well.

Following the suggestion, we revised the manuscript as follows.

“A number of miRNA detection techniques are currently commercialized, but they have their own merits and demerits.”

→

“A number of miRNA detection techniques are currently commercialized, but they have their **own**

limitations¹⁰⁻¹⁴.”

The Authors should avoid vague statements such as “extremely slow process” (line 65) and instead provide approximate numbers.

Following the suggestion, we revised the manuscript as follows.

“In miRNA detection using bare DNA probes¹⁰, the target binding of DNA probes is an extremely slow process^{11, 12}”

→

“In miRNA detection using bare DNA probes¹⁷, the target binding of DNA probes is an extremely slow process with a binding rate around 1,000 s/nM^{18,19}.”

The Authors should introduce the rationale behind using the TtAgo.

Following the suggestion, we added the following sentences to the manuscript.

“TtAgo was selected among various Argonaute proteins from different species, because it is well characterized in the laboratory, stable, and does not have a cleavage activity at the temperature used for the experiments (30 °C)^{20,21}. It is possible that a better choice of Argonaute proteins exists for Ago-FISH.”

Before the Authors use the term “duty cycle” they should first introduce it.

Following the suggestion, we revised the manuscript as follows.

“Quantitatively, the average duty cycle of the mutated region-targeting probes decreased below 0.05 compared to the original values of 0.19, 0.17, and 0.23 of let-7a_PS, let-7a_PM, and let-7a_PT, respectively (Fig. 2c).”

→

“For a quantitative analysis, we selected all time traces which showed the binding of any of the three DNA probes during observation, and obtained the average duty cycle (the fraction of observation time in which a probe is binding to the target miRNA) of each probe. Fig. 2c shows that the average duty cycle of the mutated region-targeting probes decreased significantly compared to the original values of 19%, 17%, and 23% of let-7a_PS, let-7a_PM, and let-7a_PT, respectively (Fig. 2c).”

The sentence that starts in line 97 should be corrected (the “polymerases adenosine tails” part).

The error is corrected as follows.

“We first purify whole endogenous RNAs from cell lysate, and then add adenosine tails to 3'-end of the purified RNAs using yeast poly(A) polymerase.”

In Fig. 3a there seems to be an error (Poly (X) instead of Poly (A)).

The error is corrected.

Response to the Comments of Referee #2:

Comments for the Authors

In the study by Shin and co-workers, the authors describe a method for single-molecule miRNA detection that is a variation of that reported by Johnson-Buck et al. in 2015. In the current study, miRNA probes were loaded into recombinant Thermus thermophilus Argonaute (TtAgo) and incubated with target RNAs immobilized on a polymer-coated quartz surface. Binding of miRNA probes were monitored using a multi-color single-molecule fluorescence microscope. The authors show the TtAgo-loaded miRNA probes had an increase binding rate compared to bare probes, were able to distinguish single nucleotide mutations in target miRNAs, and were able to quantify endogenous miRNA levels in total RNA from HeLa cells. Overall this is an interesting study featuring an innovative miRNA profiling technique, miRQ-ALP, that is sensitive, specific and does not require enzymatic amplification. However, additional information about experiments described in this study and the use of this technique should be provided so that this technique can be more widely useful.

We would like to thank the Reviewer for the favorable evaluation of our paper.

Specific comment/concerns:

Fig. 1. Please clarify the stoichiometry of DNA probe loaded into TtAgo. Is the ratio of probe to TtAgo the same for all probes? Please clarify whether the experiment in Fig. 1d involves incubation of the three probes with target simultaneously or separately.

Following the suggestion, we revised the Method section as follows.

“To assemble TtAgo and DNA probes, the three kinds of DNA probes (0.2 μM) was separately incubated with TtAgo (1 μM) for 30 minutes at 55 °C in an assembly buffer (10 mM Tris-HCl (pH 8.0) buffer with 100 mM NaCl, and 5 mM MgCl₂). TtAgo and DNA probes make a complex with a 1:1 stoichiometry, but excessive amount of TtAgo was used to reduce the amount of free DNA probes not loaded into TtAgo. For Ago-FISH experiments, the three kinds of TtAgo-loaded DNA probes were mixed, diluted to the final concentration of 2 nM in an imaging buffer (20 mM Tris-HCl (pH 8.0) buffer with 135 mM KCl, 0.5% formamide, 120 mM UREA, 1 mM Mg²⁺, and oxygen scavenger system: 5 mM protocatechuic acid, 500 nM protocatechuate 3,4-dioxygenase, and saturated Trolox (25 mg/50 mL)), and injected into the detection chamber.”

Please provide rationale for design of probes binding to seed, mid and tail segments of target miRNA and whether GC content and probe length was important for probe design.

Following the suggestion, we added the following sentence to the manuscript.

“When we design DNA probes, the lengths of the complementary part of the DNA probes were optimized so that they have comparable binding/dissociation kinetics on target miRNAs by compensating the varying GC contents of the seed, mid, and tail regions. The sequence of non-complementary part of DNA probes is not critical but should be selected for DNA probes not to form secondary structures. The overall length of DNA probes was maintained at 21 nt.”

Please provide data for control experiment conditions – fluorescence intensity time traces without target or in the presence of probe competitor.

Following the suggestion, new figures (Fig. 1d-f) and sentences are added to the manuscript as follows.

“Following the injection of DNA probes, a large number of single-molecule spots appeared in all the imaging channels of Cy3, Cy5, and Cy7 (Fig. 1d). On the other hand, a negligible number of single-molecule spots were observed without target miRNA immobilization (Fig. 1e), and the number of single-molecule spots were significantly reduced in the presence of ten times excessive unlabeled competitor DNA probes (Fig. 1f).”

Fig. 2b. It is not clear why intensity time traces for a probe look different for mutant targets that do not have a mutation in probe's complementary sequence. For example, the time traces of Cy7 were different for let-7a_m2 and let-7a_m3.

Binding of DNA probes is a random process, and therefore individual time traces may look different by eye inspection. For that reason, we introduced the duty cycle criteria to identify target miRNAs, and it works nicely as demonstrated in the manuscript.

The probe labeling method and experimental setup for Fig. 2e described in Supplemental Tables 1 and 2 are very confusing.

To address the comment, we replaced Supplementary Table 1-2 with Supplementary Figure 1, and revised the Method section as shown below.

“**Oligonucleotides preparation.** All oligonucleotides were purchased from Integrated DNA technology (IDT, Coralville, IA). DNA probes were labelled with Cy3, Cy5, or Cy7 mono NHS-ester which reacts with the amine group on the probe strand by incubating 0.5 mM DNA oligo with 10 mM fluorophore in a reaction buffer (100 mM Na₂BO₇ pH 8.5) for 6 hours. The excess dye was removed by using ethanol precipitation. Labelled oligonucleotides were stored in 10 mM Tris-HCl (pH 8.0) with 50 mM NaCl.”

→

“**Oligonucleotides preparation.** All oligonucleotides were purchased from Integrated DNA technology (IDT, Coralville, IA). **Amine modified** DNA probes were labelled with Cy3, Cy5, or Cy7 mono NHS-ester which reacts **with amine reactive dyes (Cy3, Cy5, or Cy7 mono NHS-ester, GE Healthcare)** by incubating 0.5 mM DNA oligos with 10 mM fluorophores in a reaction buffer (100 mM Na₂BO₇ pH 8.5) for 6 hours. The excess dyes were removed by using ethanol precipitation. Labelled oligonucleotides were stored in 10 mM Tris-HCl (pH 8.0) with 50 mM NaCl. **Information about oligo sequences is available in Supplementary Figure 1.**”

Fig. 3. No information is provided on the efficiency of the endogenous immobilization process, which required RNA isolation, addition of polyA tails, and hybridization with biotinylated polyT DNA. Please show control experiment findings for Fig. 3b. It would also be important to show that this technique can capture dynamic changes in miRNA expression, such as when cells are stimulated. How would and immobilization be normalized for comparison between different experimental conditions?

To address the comment, we added the following sentences to the manuscript.

“In Ago-FISH, the average spot numbers, and thus the quantifications of miRNAs would be affected by the efficiencies of the RNA isolation, poly(A) tailing, and hybridization of miRNAs with biotinylated poly(T) DNA. We confirmed that the efficiencies of the poly(A) tailing and hybridization were almost perfect by comparing the spot numbers observed with biotinylated let-7a and poly(A)-tailed let7a (Supplementary Fig. 9). We could not directly measure the RNA isolation efficiency, but observed the consistency of the efficiency of total RNA isolation by comparing the average let-7a spot numbers for three different batches of RNA isolation in the laboratory (17.3±3.8, 18.0±2.9, and 16.7±3.9), and a commercial one (17.0±2.0, Takara). These observations confirm that our method of

miRNA quantification is quite reliable.”

To show that this technique can capture dynamic changes in miRNA expression, we added let-7a overexpression and Drosha knockout data as Fig. 3f and Supplementary Fig. 7. We also added the following sentences to the manuscript.

“To test whether Ago-FISH can monitor the dynamic change of miRNA expression, we quantify let-7a when it is overexpressed in various cells (Method section). We found that the let-7a number per single cell increased significantly (Fig.3f, middle; Supplementary Fig. 7) when let-7a is overexpressed. On the other hand, let-7a vanished (Fig.3f, right) in Drosha knockout cells (Method section), confirming that Ago-FISH can reliably monitor the dynamic change of miRNA expression.”

In Fig. 3, it is not surprising that the slopes for the experiments in 3d are similar to those in 3e, since the levels of spike-in miRNA are much greater than endogenous miRNA. What are the slopes when <1 pM of synthetic miRNA is spiked-in, which would be comparable to endogenous levels. It is not clear that the calibration parameter of 2.8 spot/pM for quantification of miRNA levels in 3f is accurate based on curves in 3e where the lowest concentration of spike-in was 25 pM.

Following the suggestion, we repeated the experiments with lower levels of spike-in miRNA concentrations. Please see modified Fig. 3d-e.

Please clarify the amount of synthetic and spike-in miRNA in Supplementary Fig. 1.

The units were missing, and the error is corrected. The figure is renumbered as Supplementary Fig. 6 in the revision.

Details of the qRT-PCR experiment in Supplementary Fig. 2 are missing. For example, were Ct values normalized to small RNA?

Ct values are not normalized. To clarify the point, we added the following sentence to the caption of Supplementary Fig. 8 of the revised manuscript.

“Ct values are not normalized.”

We also revised the Method section as follows.

“qRT-PCR experiments for miRNA quantification were performed using TaqMan MicroRNA Assays (Applied Biosystems).”

→

“Purified total RNA was treated with DNase I (Takara). cDNAs were then synthesized using the Taqman miRNA Reverse Transcription Kit (Thermo Scientific) and subjected to quantitative real-time PCR with the Taqman MicroRNA Assay (Applied Biosystems) on StepOnePlus Real-Time PCR System (Thermo Scientific).”

Please provide additional information about data presented in Supplementary Fig. 3. Why do the fluorescent miRQ-ALP probes not bind to mature miRNA sequence in the pre-miRNA? Is this phenomenon dependent on incubation temperature?

Following the suggestion, the caption of Supplementary Fig.10 of the revised manuscript is rewritten as follows.

“None of the pre-let-7a was detected.”

→

“miRNA detection experiment was conducted at 30 °C where secondary structure of pre-miRNA is stable. Therefore, DNA probes cannot access the target region of pre-miRNA, and none of the pre-let-7a was detected.”

Profiling of miRNAs in plasma or serum would be an important application of the miRQ-ALP technique. Is this method able to detect extracellular miRNAs in plasma or serum?

Profiling of exosomal miRNAs in plasma is an important project that we should pursue in future works, but we could not find a collaborator who can provide blood samples at this stage. Instead, we quantified exosomal miRNAs from MCF-7 cell (Fig. 3g).

Response to the Comments of Referee #3:

Comments for the Authors

The article by Shin et al, describes a single-molecule miRNA detection platform that uses Ago-bound fluorescent probes. This is accomplished by visualizing Ago-probe association with immobilized target miRNAs using super resolution microscopy. The technology is specific, being able to distinguish single base pair differences, and is less fraught with artifacts when compared to qPCR methods. The authors demonstrate the specificity and sensitivity using synthetic oligos and purified total RNA. They go on to provide an exact calculation of let-7 molecules in a HeLa cell. The technology appears to be reliable platform, and the experiments demonstrating its performance are satisfying. Overall though, this is more of an evolutionary development rather than revolutionary. Demonstrating the application described at the end of manuscript (in vivo, single molecule miRNA imaging with super resolution microscopy) would likely garner more interest in the article. The authors also "over-sell" the platform in a couple ways that should addressed. The data in the manuscript are well-presented, however, I have several recommendations for the authors.

We would like to thank the Reviewer for the favorable evaluation of our paper.

The most significant change I recommend is providing additional panels for figure 3b that shows let-7c spots. It would be compelling to see distinct spots correspond to let-7a and let-7c for the tail probe upon changing the probe set.

Following the suggestion, we added a new panel to figure 3b that show let-7c spots upon probe set change.

The authors point out that qPCR suffers from enzyme-introduced artifacts. The platform presented by the article requires polyA polymerase modification of target RNAs. There are biases associated with this enzyme as well. The authors should qualify these statements, or demonstrate that there is no bias associated with polyA polymerase.

As the Reviewer pointed out, it is known that RNA primary and secondary structures influence RNA 3'-end tailing (doi:10.1093/nar/gkt1021). However, the variation of tailing efficiency due to primary structure could be overcome by long incubation time (~1 hour) (doi: 10.1261/rna.2242610). Variation of tailing efficiency due to secondary structures could be avoided by denaturing RNAs before tailing (doi:10.1093/nar/gkt1021 and doi: 10.1261/rna.2682311), too. To clarify the point, we revised the Method section as follows.

"Purified RNAs were heated at 75°C for 3 minutes to avoid biases likely caused by the aggregations and secondary structures of RNAs^{11, 31}. Yeast poly(A) polymerase (Thermo Fisher) was used to add the poly(A) tail to RNA. Specifically, 0.1 µg/µl RNA was incubated at 37 °C for 1 hour in a 20 mM Tris-HCl (pH 7.0) buffer with 20 µM EDTA, 0.2 mM DTT, 0.6 mM MnCl₂, 10% glycerol, 100 µg/ml acetylated BSA, 2.5 mM rATP, 40 U/µl yeast poly(A) polymerase (Thermo Fisher), and 0.5 U/µl RNase inhibitor (SUPERase.InTM, Thermo Fisher). The reaction was terminated by heating at 65°C for 15 minutes. It was reported that the 3'-terminal nucleotide influences the poly(A) tail efficiency¹¹. In our method, the capturing efficiency is not expected to vary significantly once the poly(A) tail is longer than 30nt (the length of the complementary DNA use for capturing miRNAs). Furthermore, the influences of the 3'-terminal nucleotide on miRNA capturing efficiency could be effectively avoided by maintaining the poly(A) tailing reaction for 1 hour³²."

It is unclear why the authors focus exclusively on microRNAs. It would seem that any nucleic acid sequence could be targeted by the Ago probes. TtAgo is a DNA binding Ago and does not target RNAs

in vivo. Seems that binding could occur on any immobilized nucleic acid. This opens many applications that would benefit from the enhanced, prolonged, specific binding of the Ago-probes.

Thank you for the suggestion. Following the suggestion, we added following sentences to Discussion.

“In this work, Ago-FISH is used only to quantify miRNAs. The same technique, however, can be utilized to quantify other nucleic acids such as mRNAs with accelerated binding rate, improved specificity, and not limited by photobleaching.”

The authors initially describe the platform as having implications for the clinic. While the target binding aspect of the platform can certainly be miniaturized to a “chip”, the super resolution microscopy needed to read the chip is not going to be a “rapid” option due to equipment being unavailable outside the most well-equipped medical center.

Single-molecule imaging instruments can be miniaturized to a table-top box size (215mm X 420mm X 250mm; Oxford Nonoimaging:<https://oni.bio/>). We believe the price of the instrument can be reduced significantly when the demand for the instrument increases.

Reviewers' Comments:

Reviewer #1:

Remarks to the Author:

I thank the Authors for addressing all of my comments in a highly satisfactory manner. The manuscript in its current form is rich in detail, very clear and most importantly seems to describe a very relevant new method for what appears to be a truly sensitive and specific detection of miRNAs (and possibly beyond).

I still have very minor comments and I would appreciate if the Authors addressed them:

1. Please rephrase "normal people" to "healthy individuals".

2. Please rephrase the following sentence:

"The accelerated target binding of TtAgo-loaded DNA probes will allow the combination of Ago-FISH with superresolution fluorescence microscopy of endogenous miRNAs, which will provide the valuable information on the spatial distribution of miRNAs in cells and tissues."

To

"The accelerated target binding of TtAgo-loaded DNA probes will allow the combination of Ago-FISH with superresolution fluorescence microscopy of endogenous miRNAs, which might provide valuable information on the spatial distribution of miRNAs in cells and tissues."

...given that this has not yet been demonstrated.

3. In general, throughout the text, there seems to be a mix of tenses even in the same sentence. This should be corrected so that the use of tense is more consistent.

Reviewer #2:

Remarks to the Author:

In the revised version of their manuscript, Hohng and co-workers addressed reviewers concerns about experimental detail, probe design, assay conditions, data analysis, and comparison of their technique to current gold standard techniques. They have also renamed their technique Ago-FISH and included additional data showing that their method can distinguish the miRNAs with different 3'-tail lengths. A key point of clarification is the role of average duty cycle in the quantitative analysis. Overall the manuscript is improved. However, a few questions remain that would be of importance to investigators who are interested in using this technique.

Specific comments/questions:

1. The authors acknowledge a wide variation in imaging times for different experiments. Please clarify how imaging time is determined for a given experiment. Does this depend on target miRNA so that imaging time would have to be optimized for each miRNA?

2. Please clarify at what point in the experiment are single-molecule spots imaged.

3. For the experiment in Figure 2, please clarify why duty cycle changes for probes that are not supposed to be affected by mutation (e.g. duty cycle for PT in let-7a_m2 experiment). Is this important? Could this be because PT probe appears to overlap with mutation in mid segment region?

4. Supplemental figures 1 and 2 legend should contain more information about what is being

shown.

5. In Figure 3, why would one expect to see similar spot/pM slopes for other miRNAs, besides let-7a and let-7c? It is known that qRT-PCR efficiency and specificity is not uniform for all miRNAs; is it possible that Ago-FISH efficiency and specificity might show similar variability, depending on miRNA?

6. Please clarify why no statistical analysis has been performed for quantitative analyses.

Reviewer #3:

Remarks to the Author:

The authors have satisfied most of my recommendations. I believe the manuscript is suitable for publication.

Response to the Comments of Referee #1:

Comments for the Authors

I thank the Authors for addressing all of my comments in a highly satisfactory manner. The manuscript in its current form is rich in detail, very clear and most importantly seems to describe a very relevant new method for what appears to be a truly sensitive and specific detection of miRNAs (and possibly beyond). I still have very minor comments and I would appreciate if the Authors addressed them:

We would like to thank the Reviewer for the favorable evaluation of our paper, and we addressed all the comments of the Reviewer as shown below.

1. Please rephrase "normal people" to "healthy individuals".

Revised as suggested.

2. Please rephrase the following sentence:

"The accelerated target binding of TtAgo-loaded DNA probes will allow the combination of Ago-FISH with superresolution fluorescence microscopy of endogenous miRNAs, which will provide the valuable information on the spatial distribution of miRNAs in cells and tissues."

To

"The accelerated target binding of TtAgo-loaded DNA probes will allow the combination of Ago-FISH with superresolution fluorescence microscopy of endogenous miRNAs, which might provide valuable information on the spatial distribution of miRNAs in cells and tissues."

...given that this has not yet been demonstrated.

Revised as suggested.

3. In general, throughout the text, there seems to be a mix of tenses even in the same sentence. This should be corrected so that the use of tense is more consistent.

We read the manuscript carefully and corrected all the grammatical errors that we found.

Response to the Comments of Referee #2:

Comments for the Authors

In the revised version of their manuscript, Hohng and co-workers addressed reviewers concerns about experimental detail, probe design, assay conditions, data analysis, and comparison of their technique to current gold standard techniques. They have also renamed their technique Ago-FISH and included additional data showing that their method can distinguish the miRNAs with different 3'-tail lengths. A key point of clarification is the role of average duty cycle in the quantitative analysis. Overall the manuscript is improved. However, a few questions remain that would be of importance to investigators who are interested in using this technique.

We would like to thank the Reviewer for the favorable evaluation of our paper, and addressed all the comments of the Reviewer as shown below.

Specific comments/questions:

1. The authors acknowledge a wide variation in imaging times for different experiments. Please clarify how imaging time is determined for a given experiment. Does this depend on target miRNA so that imaging time would have to be optimized for each miRNA?

We made a wrong impression in the point-by-point responses of the first revision that we had used widely varying imaging times for miRNA detection. In reality, we consistently used 20 min observation time for statistical analysis of miRNA detection. Different observation times were used only for other purposes. To be specific, 10 min imaging time was used for the determination of k_{on} and k_{off} of Ago-loaded DNA probes, and 1 hr imaging time was used for the determination of k_{on} and k_{off} of bare DNA probes (Fig. 1h-l). For all other experiments of miRNA detection, 20 min imaging time was consistently used except the rare cases of experiments to obtain time traces (Fig. 1g, 2b). To clarify the point, we added the information concerning the observation times in the 2nd revision.

2. Please clarify at what point in the experiment are single-molecule spots imaged.

Single-molecule imaging was started right after the injection of DNA probes. To clarify the point, we revised the manuscript as follows.

“Experiments were conducted at 30 °C.”

→

“Single-molecule fluorescence imaging experiments were started right after the injection of DNA probes at 30 °C.”

3. For the experiment in Figure 2, please clarify why duty cycle changes for probes that are not supposed to be affected by mutation (e.g. duty cycle for PT in let-7a_m2 experiment). Is this important? Could this be because PT probe appears to overlap with mutation in mid segment region?

The reviewer’s reasoning for the duty cycle drop of PT probe in let-7a_m2 target is correct. To clarify the point, we added the following sentences to the revision.

“It was noticeable that the duty cycle of let-7a_PT was also reduced a little in the case of let-7a_m2 because the 3'-end of the target recognition sequence of let-7a_PT probe overlapped the mutated base of let-7a_m2. This reduction, however, did not cause any problem but only favorable effect in target discrimination.”

4. Supplemental figures 1 and 2 legend should contain more information about what is being shown.

Following the suggestion, we added more descriptions to the figure captions.

5. In Figure 3, why would one expect to see similar spot/pM slopes for other miRNAs, besides let-7a and let-7c? It is known that qRT-PCR efficiency and specificity is not uniform for all miRNAs; is it possible that Ago-FISH efficiency and specificity might show similar variability, depending on miRNA?

In our scheme of miRNA detection, all different miRNAs are captured using the same poly(A) tail, and therefore similar capturing efficiencies of different miRNAs are naturally expected. However, the miRNA detection efficiencies of DNA probes may change for different miRNAs. To clarify the point, we added the following sentences to the revision.

“In our scheme of miRNA detection, all different miRNAs are captured using the same poly(A) tail, and therefore similar capturing efficiencies of different miRNAs are naturally expected. However, the miRNA detection efficiencies of DNA probes depend on the design of DNA probes. It remains to be tested whether the similar calibration parameters can be obtained for other miRNA targets by optimizing DNA probes.”

6. Please clarify why no statistical analysis has been performed for quantitative analyses.

I am afraid that we cannot understand the true meaning of the Reviewer's comment because we already extensively used statistical analysis in the manuscript. To make the paper more quantitative, however, R^2 values of data fitting curves are additionally included in the revision.

Response to the Comments of Referee #3:

Comments for the Authors

The authors have satisfied most of my recommendations. I believe the manuscript is suitable for publication.

We would like to thank the Reviewer for the favorable evaluation of our paper.

Reviewers' Comments:

Reviewer #2:

Remarks to the Author:

Thanks to the authors for their thoughtful responses to reviewer comments. The authors have convincingly shown that their method is sensitive, specific and significantly faster than current gold standard techniques for miRNA detection and quantification. Assuming the that a single-molecular fluorescence microscope is readily available, the most time-consuming part of this method appears to be the design of specific probes for each miRNA that needs to be quantified. However, this design process would have to be done for any miRNA detection method. The previous comment about statistical analysis refers to figures in which mean data +/- SD are shown – it is not clear whether statistical analysis (i.e. p values) has been performed when different mean values are compared. This is most obvious for figures 3f, g, h, where the average copy numbers are shown for different cell types, tissues, exosomes.

Response to the Comments of Referee #2:

Thanks to the authors for their thoughtful responses to reviewer comments. The authors have convincingly shown that their method is sensitive, specific and significantly faster than current gold standard techniques for miRNA detection and quantification. Assuming the that a single-molecular fluorescence microscope is readily available, the most time-consuming part of this method appears to be the design of specific probes for each miRNA that needs to be quantified. However, this design process would have to be done for any miRNA detection method. The previous comment about statistical analysis refers to figures in which mean data +/- SD are shown – it is not clear whether statistical analysis (i.e. p values) has been performed when different mean values are compared. This is most obvious for figures 3f, g, h, where the average copy numbers are shown for different cell types, tissues, exosomes.

The manuscript and figures are revised as suggested.